# Specific Glutamylation Patterns of the Cytoskeleton Confer Neuroresistance to Lobe X of the Cerebellum in a Model of Childhood-Onset Neurodegeneration with Cerebellar Atrophy

**DOI:** 10.3390/ijms262110378

**Published:** 2025-10-25

**Authors:** Carlos Hernández-Pérez, Andrés A. Calderón-García, David Pérez-Boyero, Verónica González-Núñez, Eduardo Weruaga, David Díaz

**Affiliations:** 1Laboratory of Neuronal Plasticity and Neurorepair, Institute for Neuroscience of Castilla y León (INCyL), Universidad de Salamanca, C/Pintor Fernando Gallego 1, E-37007 Salamanca, Spain; carlosh@usal.es (C.H.-P.); dpb@usal.es (D.P.-B.); 2Department of Anatomy and Radiology, Universidad de Valladolid, Av. Ramón y Cajal 7, E-47005 Valladolid, Spain; andresangel.calderon@uva.es; 3Institute of Biomedical Research of Salamanca (IBSAL), P.º de San Vicente, 182, E-37007 Salamanca, Spain; vgnunez@usal.es; 4Department Biochemistry and Molecular Biology, Faculty of Medicine, Universidad de Salamanca, C/Alfonso X el Sabio, s/n, E-37007 Salamanca, Spain; 5Laboratory of Neurochemistry and Molecular Neuroscience, Institute for Neuroscience of Castilla y León (INCyL), Universidad de Salamanca, C/Pintor Fernando Gallego 1, E-37007 Salamanca, Spain

**Keywords:** CCP, cerebellum, CONDCA, glutamylation, Purkinje cell degeneration, neuroprotection

## Abstract

The cytoskeleton relies heavily on the dynamic nature of microtubules, regulated by post-translational modifications such as polyglutamylation and deglutamylation. Disruption of its internal balance, particularly through the absence of cytosolic carboxypeptidase 1 (CCP1), leads to cytoskeletal collapse and cell death. An example of this occurrence exists in the Purkinje Cell Degeneration (PCD) mouse, a direct animal model for childhood-onset neurodegeneration with cerebellar atrophy (CONDCA) human disease. Both CONDCA patients and PCD mice suffer a dramatic degeneration of Purkinje cells. Intriguingly, lobe X appears less vulnerable to this insult. This study revealed in wild-type mice that lobe X expresses less *Ccp1* compared to other lobes, correlating with its delayed degeneration in PCD mice. Further expression analysis of other deglutamylating enzymes (CCP4 and CCP6) and glutamylating enzymes (TTLL1) revealed distinctive patterns: *Ccp4* showed minimal relevance in cerebellum, while *Ccp6* displayed a compensatory increase during critical stages. Meanwhile, *Ttll1* expression remained consistent across lobes, suggesting that the resistance of lobe X may be related to a more dynamic, hyperglutamylated cytoskeleton. Unraveling the neuroresistance mechanisms of Purkinje cells may help mitigate neuronal loss in CONDCA patients and may offer a glimmer of hope for alleviating the symptoms of other neurodegenerative diseases.

## 1. Introduction

The Purkinje Cell Degeneration (PCD) mouse is a model of cerebellar degeneration caused by an autosomal mutation in the *Ccp1* gene, leading to a progressive Purkinje cell loss and severe ataxia. This mutant is considered a direct genetic model of human childhood-onset neurodegeneration with cerebellar atrophy (CONDCA), as children with biallelic *CCP1* mutations display similar pathophysiological and clinical features [1,2,3]. In the PCD cerebellum, two stages can be distinguished: a pre-degenerative phase (from postnatal day 15-P15-to P18), characterized by nuclear, cytological, and morphological changes in surviving Purkinje cells, followed by a degenerative phase starting at P18, when neuronal death accelerates and nearly all Purkinje cells are lost by P35 [4]. In humans, these two stages have not been already demonstrated, probably due to the difficulties in analyzing human tissue, especially derived from children and in prodromal stages of a neurodegenerative process. Although the PCD mouse is primarily a model of Purkinje cell degeneration, other cell types, such as olfactory bulb mitral cells, also undergo milder degeneration [5].

The *Ccp1* gene encodes a peptidase known as cytosolic carboxypeptidase type 1 (CCP1). This enzyme hydrolyzes the carboxy-terminal glutamate side chains of tubulins [6]. To understand the importance of this gene on Purkinje cells, it is necessary to consider the post-translational modifications of their microtubules. Microtubules consist of α and β tubulins [7] and are highly dynamic due to modifications such as polyglutamylation and deglutamylation [8].

Polyglutamylation is a reversible modification in which glutamate side chains are added to tubulins by tubulin tyrosine ligase-like enzymes or TTLLs [9]. Nine TTLLs are involved in this process [10]. Several models have shown that both blocking and overexpressing TTLLs cause severe cytoskeletal defects [9,11,12]. Among them, TTLL1 is particularly relevant: mice lacking TTLL1 develop abnormal cilia in the respiratory epithelium that lead to breathing problems [11].

Deglutamylation is the reverse process of polyglutamylation in which cytosolic carboxypeptidases (CCPs) remove the glutamates previously added by TTLLs. CCP1, and to a lesser extent CCP4 and CCP6, shorten glutamate side chains [6], while CCP5 removes the initial glutamate of each side chain [13]. Knockout models of CCPs show distinct alterations: loss of CCP1 produces excessive polyglutamylation similarly to TTLL1 overexpression, and both conditions cause the death of Purkinje cells [14].

Polyglutamylation rises sharply in the cerebral cortex and cerebellum during the early development. After that, in wild-type mice, *Ccp1* expression increases significantly from P15, reducing glutamylation and stabilizing the cerebellar cytoskeleton [4,15]. By contrast, this upregulation is absent in PCD mice, leading to cytoskeletal hyperglutamylation and subsequent Purkinje cell degeneration due to insufficient CCP1 [4,6,15]. However, glutamylation returns to normal levels in the cerebral cortex of PCD mice [6], suggesting a compensatory mechanism, likely mediated by CCP6. Supporting this idea, in situ hybridization has shown that CCP6 expression is mainly restricted to the cerebral cortex [16]. Although *pcd* mutation is considered recessive, Northern blot analyses allowed the detection of a *Ccp1* mRNA residual expression in testis [17,18]. These findings indicate that PCD mice can still express the gene, but at levels too low to maintain a stable cytoskeleton in different neuronal populations.

Quantitative PCR (qPCR) analyses of wild-type mice allowed the analyses of normal expression temporal patterns of *Ccp1* in the cerebellum and olfactory bulb [19]. In the cerebellum, expression peaks significantly between P15 and P25, whereas in the olfactory bulb expression remains low at P60-P70, with only a slight increase. These data are particularly revealing when compared to the timeline of neuronal degeneration in PCD mice: Purkinje cell death begins around P20 [4], while neuronal loss in the olfactory bulb starts later, at P70 [5,20], the bulbar degeneration being less severe and occurring more slowly than the former cell population. These findings suggest tissue-specific dependence on CCP1. However, cerebellar expression has only been studied up to P25, leaving open the question of its dynamics at later stages when Purkinje cell loss continues in PCD mice.

Here, it is important to note that lobe X has a particular neuroresistance not only in the PCD mouse, but also in other several models such as *Leaner*, *Toppler*, *Robotic*, *Shaker*, *Lurcher*, NPC1 or *Nervous* [21]. Understanding the mechanisms that underlie this selective protection compared to the rest of the vermis may provide insight into cerebellar degeneration and identify potential therapeutic targets.

We have previously demonstrated a higher basal expression of Heat Shock Protein 25 (HSP25), its phosphorylated active form and its related kinase in lobe X, along with an additional upregulation in response to neuronal. Since HSP25 confers protection against diverse cellular stressors, these findings suggest that lobe X is inherently more protected than other cerebellar regions against any type of neural damage. Moreover, apart from intrinsic molecular factors, such as HSP25 expression, it cannot be excluded that the particular vascularization, blood–brain barrier permeability, or trophic environment of lobe X may also contribute to its resilience. Enhanced access to neurotrophic factors and supportive glial interactions might provide an additional protective background for Purkinje cells in this region (see Section 3, Discussion).

Beyond this constitutive neuroprotection, more specific mechanisms may contribute to the relative resistance of lobe X in the PCD model. Then, this work focuses on a possible additional resistance of lobe X of the cerebellum in the PCD model.

As previously stated, *Ccp1* expression in wild-type mice coincides temporally, spatially and in intensity with the pattern of Purkinje cell degeneration in PCD mice. We therefore hypothesized that lobe X might exhibit lower *Ccp1* expression than other lobes, making it less dependent on *Ccp1* and thus less vulnerable to the *pcd* mutation. Conversely, compensatory increases in *Ccp4* and *Ccp6* or reduced *Ttll1* may mitigate the absence of *Ccp1* [14,16]

Accordingly, we analyzed the expression of *Ccp1*, *Ccp4, Ccp6* and *Ttll1* in wild-type mice from P20 to P50, a critical window for PCD cerebellar degeneration, comparing lobe X with the remaining lobes. We also examined *Ccp4*, *Ccp6* and *Ttll1* in PCD tissues to identify potential compensatory mechanisms.

Whereas our parallel work (see explanation above) highlights a physiological feature of lobe X that confers it an **innate protection against any neuronal damage**, the present study addresses the **decreased vulnerability** of lobe X to the *pcd* mutation. While both contribute to the neuroresistance observed in lobe X of PCD mice, they have distinct underlying causes.

## 2. Results

### 2.1. Temporal Dynamics of Ccp1, 4 and 6, and Ttll1 Gene Expression in Different Cerebellar Lobes of Wild-Type Mice

The results obtained for the temporal variation in gene expression in lobe X were different depending on the gene being studied. For example, *Ccp1* expression changed over time (*p* < 0.001; Figure 1A; Appendix A). Using *post hoc* test, we were able to group the ages into the statistically most likely subsets. By doing so, P20, P35 and P40 constituted one group, P25 and P30 another, and, finally, P50 constituted a third individual group (Figure 1A). The experimental results showed that the expression of *Ccp1* dropped from P20 to P25 and was maintained until P30. At P35, *Ccp1* expression again increased to its initial level and was maintained until, at least, P40. After P40, expression dropped until P50, reaching half the minimum observed between P25-P30 and the previous level of P35–P40. Regarding *Ccp4*, its expression in lobe X did not significantly change among the age groups (Figure 1B; Appendix A). Lastly, *Ccp6* and *Ttll1* presented similar statistical results: the Kruskal–Wallis test confirmed significant differences in both gene expression patterns (*p_Ccp6_* = 0.006 and *p_Ttll1_* = 0.014). Also, the *post hoc* tests grouped all ages into a single group, except for P30, which constituted an individual group (Figure 1C,D; Appendix A). Both genes behaved similarly as their expression at P20 was maintained until P25. This expression decreased at P30 and then returned to the initial levels in the rest of the age groups (Figure 1C,D).

In addition to lobe X, the expression patterns of the same 4 genes mentioned above were examined in lobes I–IX throughout the different age groups. In the case of *Ccp1*, the pattern observed was similar to that observed in lobe X, with significant temporal differences (*p* = 0.001; Figure 1E): the initial expression detected at P20 dropped at P25 and then again at P30. From this age and onward, the expression of *Ccp1* increased to its original levels and was maintained (Figure 1E; Appendix A). Regarding the expression of *Ccp4* in lobes I–IX, significant differences were observed (*p* = 0.041; Figure 1F) and the ages were grouped into an initial group comprising ages from P20 to P40 and a final group at P50, when *Ccp4* expression was slightly higher (Figure 1F; Appendix A). However, these data must be considered with caution as the level of expression detected was extremely low (the cycle threshold or “C_T_” of qPCR for this gene was always > 30; see Discussion below). Significant differences were also observed for the expression of *Ccp6* (*p* = 0.002 Figure 1G; Appendix A). In addition, the age groups were arranged in the following pattern: the initial expression at P20-P25 decreased at P30 and then increased at P35 and P40, and P50 again appeared as an independent group with the highest level of expression (Figure 1G). Finally, the expression pattern of *Ttll1* also showed significant differences (*p* = 0.009; Figure 1H) and the same pattern observed in lobe X: all ages made up a single group except P30, which showed a lower expression than the rest (Figure 1H; Appendix A).

### 2.2. Genes Ccp1, 4 and 6 and Ttll1 Exhibit Different Expression Patterns in Lobe X Compared to the Rest of the Lobes

The individual expression patterns of genes *Ccp1*, *Ccp4*, *Ccp6* and *Ttll1* in wild-type mice in lobe X were compared to those found in the rest of the lobes at ages P20, P25, P30, P35, P40 and P50 separately.

At P20, in lobe X, *Ccp1* and *Ccp4* expression in lobe X was lower than in the rest of the lobes (*p_Ccp1_* = 0.008 and *p_Ccp4_* = 0.008; Figure 2, Appendix A). Conversely, *Ccp6* and *Ttll1* did not show significant differences between either region (Figure 2, Appendix A). The results at P25 were the same as those found at P20, when *Ccp1* and *Ccp4* had a lower expression in lobe X (*p_Ccp1_* = 0.002 and *p_Ccp4_* = 0.002; Figure 2, Appendix A), and no differences were found between the regions for *Ccp6* and *Ttll1* (Figure 2, Appendix A). It should be noted that these levels of gene expression, in the wild-type mouse, suggest that lobe X constitutively requires lower levels of deglutamylation than the rest of the lobes at these two initial ages.

At P30, as in the previous ages, *Ccp1* expression was still lower in lobe X than in the rest of the lobes (*p* = 0.008; Figure 2, Appendix A), but *Ccp4* increased its expression in lobe X, thus being similar in both regions (Figure 2, Appendix A). Conversely, it is striking that *Ccp6*, which had not presented any differences before, had a greater level of expression in lobe X in relation to the rest of the lobes (*p* = 0.029; Figure 2, Appendix A). Regarding *Ttll1*, no differences between cerebellar regions were detected (Figure 2, Appendix A).

At P35, *Ccp1* continued to be less expressed in lobe X than in the rest of the lobes (*p* = 0.008; Figure 2, Appendix A). *Ccp4* and *Ccp6* did not show significant differences (Figure 2, Appendix A) and *Ttll1* had a reduced level of expression in lobe X as compared to the rest of the lobes (*p* = 0.008; Figure 2, Appendix A).

At P40, except for *Ccp1*, whose expression was again reduced in lobe X compared to the rest of the lobes (*p* = 0.002; Figure 2, Appendix A), no significant differences were found for the other the genes between the two cerebellar locations (Figure 2, Appendix A–T).

Finally, at P50, highly significant differences were observed for all genes (*p_Ccp1_* = 0.002, *p_Ccp4_* = 0.008, *p_Ccp6_* = 0.002 and *p_Ttll1_* = 0.002; Figure 2, Appendix A–X), where in all cases their expression was lower in lobe X.

These results, which correspond to wild-type animals, could be summarized as follows. First, the expression of *Ccp1* at all of the ages analyzed is lower in lobe X than in the rest of the vermis (Figure 2A, Appendix A). A possible explanation for this remarkable difference is that the cytoskeleton of the Purkinje cells of lobe X does not require the same level of deglutamylation as in the rest of the lobes; thus, there is less constitutive expression of *Ccp1* (see Section 3, Discussion). Conversely, for the rest of the genes, no general temporal pattern was observed, and differences between cerebellar regions could only be observed at specific moments (Figure 2, Appendix A).

### 2.3. Expression Patterns of Genes Ccp4, Ccp6 and Ttll1 in Lobe X vs. The Rest of the Lobes in PCD Mice

Following the analysis of gene expression patterns in wild-type mice, we focused our study on whether these differences persisted in PCD mice. The goal was to determine whether the lack of *Ccp1* could induce changes or compensatory mechanisms in the other related genes. These analyses were performed specifically at P20 and P25. We omitted later ages due to advanced degeneration of Purkinje cells in older PCD mice. Consequently, the gene expression analyses in older mice may not accurately reflect the contribution of these cells, which could have been biased by neuronal loss. Additionally, it is important to highlight that *Ccp1* expression was not included in our analysis of PCD mice, as the *pcd* mutation prevents its expression; any signal would reflect non-protein coding RNA fragments.

At P20, we verified that the pattern of gene expression in lobe X as compared to the rest of the vermis in PCD mice was similar to the results obtained in wild-type animals: a decreased expression of *Ccp4* (*p* = 0.029) and no differences for *Ccp6* and *Ttll1* (Figure 3A–C; Appendix A). Thus, the lack of *Ccp1* (*pcd* mutation) does not seem to affect the expression of the other genes at this age, both in lobe X and in the rest of the lobes. Taken together, our results indicate that lobe X requires less deglutamylation even in the background of the *pcd* mutation.

Conversely, at P25, no significant differences between cerebellar regions were found in the tissue of PCD mice (Figure 3D–F; Appendix A), also comprising *Ccp4,* which differs from the results obtained in wild-type mice (lower expression of *Ccp4* in lobe X, see before; Figure 2B; Appendix A).

### 2.4. Impact of the Pcd Mutation on Ccp4, Ccp6 and Ttll1 Expression in the Cerebellum of PCD Mice

To confirm whether the *pcd* mutation affected the expression of *Ccp4*, *Ccp6* and *Ttll1*, the fold changes in these genes were calculated by comparing their expression in lobe X and the rest of the lobes in PCD mice with that of wild-type, both at P20 and at P25 (Figure 4).

At P20, no significant differences were observed between PCD and wild-type mice regarding the expression of any of these genes, either in lobe X or the rest of the lobes. These results suggest that, at this age, the *pcd* mutation does not yet have a discernible impact on the expression of these genes in the cerebellum (Figure 4A–F; Appendix A).

At P25, no differences in gene expression in lobe X were found between PCD and wild-type mice (Figure 4G–I; Appendix A) or for genes *Ccp4* and *Ttll1* in lobes I-IX (Figure 4J,L; Appendix A). Conversely, we found that in lobes I-IX in PCD mice, *Ccp6* expression was almost half of that of wild-type mice (*p* = 0.009; Figure 4K; Appendix A). This reduction could imply that the *pcd* mutation affects *Ccp6* expression in this region and at this age. However, it is necessary to note that the results obtained for lobes I-IX should be considered with caution because at this age, the vermis region, excluding lobe X, suffers considerable neuronal loss in PCD mice. Therefore, these data may not be fully accurate due to degeneration.

### 2.5. Validation of Protein Expression

To confirm the results obtained from qPCR, we performed Western blot analyses to examine protein levels, as mRNA levels do not necessarily correspond to protein abundance. At P20, we observed the presence of CCP1 in wild-type animals and its absence in PCD animals in all cerebellar lobes (Figure 5A; Appendix A’). CCP6 and TTLL1 appeared in both tissues and genotypes (Figure 5A; Appendix A’). Our results were similar at P25: CCP1, CCP6 and TTLL1 were present in wild-type mice, while only CCP6 and TTLL1 were detected in PCD mice (Figure 5B; Appendix A). Note that GAPDH band of PCD mice at P25 has a lower intensity than the wild-type one due to the ongoing process of cell death, which also compromises general protein expression.

The advanced degeneration of Purkinje cells in PCD mice at later ages would skew the results obtained for gene and protein expression in these neurons. Therefore, the protein analysis at P30, P35, P40 and P50 was only performed using wild-type mice. In these animals, we again confirmed the presence of CCP1, CCP6 and TTLL1 in both lobe X and the rest of the vermis (Figure 6; Appendix A). Also, it is important to note that CCP4 expression was not detected at any age, genotype or tissue, which is consistent with the qPCR results detecting only extremely low levels of *Ccp4* gene expression (C_T_ for *Ccp4* was always >30).

Additionally, we performed Western blot analyses using multiple exposures (see Section 4, Materials and Methods). In general, some of the images captured with longer exposure times become overexposed and are often discarded. However, it deserves to note that overexposed images can reveal very low expression levels. This is the case of the location corresponding to the CCP1 band in PCD animals (Appendix A). This suggests CCP1 is produced in the cerebellum of PCD mice at both P20 and P25 but at extremely low levels.

Finally, we performed an image quantification of the intensity of the Western blot signal, after normalization with GAPDH. No clear general tendencies could be observed for any protein, due to the intrinsic variability of the technique and the limited number of samples (*n* = 1 or *n* = 2, as they were just confirmative). Therefore, we restricted our analyses to the ages of P20 and P25, when both experimental groups (wild type and PCD) can be compared, and for CCP1, the main protein of interest of this work. Our results demonstrated that CCP1 expression was always reduced in lobe X in comparison with the rest of the cerebellum of wild-type mice, and had scarce/residual expression in PCD animals. These results perfectly fit with our qPCR analysis, even taking into account the limitations of this semi-quantitative measure after Western blotting.

Since results for *Ccp1/*CCP1 were coherent, we decided to perform a statistical three-way ANOVA analysis for protein expression data, to compare region (lobe X vs. lobes I-IX), genotype (wild type vs. PCD) and timing (P20 vs. P25). Such analysis revealed differences in region (Figure 7; *p* = 0.0126) and genotype (Figure 7; *p* < 0.0001), thus confirming our hypothesis of a lower dependence of CCP1 in lobe X, as well as the effect of *pcd* mutation concerning this protein.

## 3. Discussion

A central result of this work is the lower expression of *Ccp1* in lobe X compared to the rest of the lobes in wild-type mice at all the ages studied. Previous work showed that Purkinje cell degeneration in PCD mice coincides in intensity and temporally with *Ccp1* expression in wild-type animals when comparing cerebellum and olfactory bulb [19]. Our results therefore suggest that vermis is more dependent on *Ccp1* than lobe X, explaining why its Purkinje cells degenerate earlier and more severely in the mutant. Although this reduced dependence delays degeneration in lobe X, it is not sufficient to prevent it entirely.

Importantly, we verified that even in wild-type animals, lobe X shows low but detectable CCP1 protein expression. In PCD mice, Western blot analyses confirmed the near absence of CCP1, although overexposure revealed a faint band consistent with residual expression. This agrees with reports of very low *Ccp1* mRNA levels in some tissues of PCD mice [17,18]. Such residual expression, however, appears too low to maintain neuronal integrity and likely contributes to the eventual degeneration of lobe X.

The temporal variations in Ccp1 expression are summarized in Figure 8, which combines our qPCR results from lobe X and the rest of the lobes with previously published data reported by [19]. The stages of cerebellar degeneration in PCD mice are also indicated: the pre-degenerative phase (P15–P18), the degenerative phase (P18–P30) [4], and the beginning of the hypothetical degenerative stage of lobe X from P30 onward. It is important to note that gene expression values for lobe X, the rest of the lobes, and the whole cerebellum cannot be directly compared, since they are relative to each region. However, as *Ccp1* expression was consistently lower in lobe X across all ages studied, its curve was plotted below that of the rest of the lobes, for clarity (Figure 8).

Baltanás et al. reported a peak of *Ccp1* expression in the cerebellum between P15–P25, followed by a lack of data until P50 (Figure 8) [19]. Our results complement these findings, showing that in both lobe X and the other lobes, expression decreases from P20 to P30, consistent with the published data. In the previously unreported interval (P25–P50), we observed a new increase at P35, followed by a slight decline at P50 in lobe X (Figure 1A). Relating these data to the stages of PCD, the first decline at P25-P30 coincides with the peak of vermis neurodegeneration, while lobe X remains morphologically preserved. This suggests that the reduced requirement for *Ccp1* during this window is tolerated in lobe X due to its constitutively lower expression (see before). Conversely, the second increase at P35 coincides with the onset of Purkinje cell loss in lobe X of PCD mice, indicating renewed dependence of *Ccp1* at this stage. This physiological rise in *Ccp1* expression around P35 may reflect an additional phase of Purkinje cell maturation and cytoskeletal stabilization, consistent with previous evidence linking *Ccp1* activity to dendritic remodeling during postnatal cerebellar development [4]. This apparent discrepancy with Baltanás et al. [19] likely reflects the fact that their study did not include intermediate ages between P25 and P50, a period in which our data reveal the transient increase in expression. Thus, both datasets are compatible when considering the different temporal sampling. Together, these findings support a temporal link between *Ccp1* expression in wild-type mice and the vulnerability of lobe X in the PCD mutant.

In a parallel work, we have demonstrated that HSP25 is more expressed in the lobe X than in the rest of cerebellum, and this expression was also higher in PCD than in wild-type mice (Hernández-Pérez et al., submitted to this same journal). HSP25 is known for its protective properties against different types of cellular stress, and we have demonstrated that its expression increases due to the neuronal loss of PCD mice: differences amongst genotypes are particularly evident from P25 onwards, which also suggests a preventive/compensatory effect. Indeed, virtually all the survival Purkinje cells of lobe X present HSP25, especially at latter ages. Moreover, the HSP25 phosphorylated active form (HSP25-P-Ser15) was almost exclusively relegated to lobe X of PCD mice. Its expression also presented differences between genotypes, being almost inexistent in wild-type animals, but increasing dramatically in PCD mice due to neuronal death. PKC-δ, the specific kinase that triggers HSP25 activation, was responsible for these changes, thus validating this putative neuroprotective pathway.

Consequently, lobe X may be intrinsically protected and therefore more resistant to CCP1 deficiency. Beyond this basal protection, additional neuroresistance mechanisms involving CCPs and related proteins may contribute to its unique vulnerability profile (see at the end of this manuscript) [21].

In addition to intrinsic mechanisms such as HSP25 expression, extrinsic factors may also contribute to the selective resistance of lobe X. This region, part of the phylogenetically ancient vestibulocerebellum, displays distinctive vascularization and metabolic profiles that may influence its susceptibility to damage. Regional differences in blood–brain barrier permeability have been described, with the cerebellum showing higher permeability than cortical areas under certain physiological and experimental conditions [22]. Such variations may facilitate the selective entry of nutrients and trophic factors, contributing to local protection. In this context, neurotrophic molecules such as GDNF, and IGF-1 have been shown to promote Purkinje cell survival and to delay neurodegeneration and motor deficits in animal models of hereditary cerebellar ataxia [23]. The combined or sustained action of these factors, together with glial cells that modulate the local microenvironment and inflammatory responses, could provide an additional protective environment. Indeed, cerebellar microglia can adopt anti-inflammatory and tissue-repair phenotypes that contribute to neuronal preservation and a less neurotoxic environment [24]. Together, these vascular, trophic, and glial influences may cooperate with intrinsic mechanisms such as HSP25 activation to explain the exceptional neuroresistance of lobe X.

The relationship between tissue vulnerability and dependence on specific CCPs has been established. CCP1, CCP5 and CCP6 are widely distributed in the brain, whereas CCP4 is mainly expressed in the eyes with minimal brain expression [16]. Accordingly, *Ccp4* mutations cause Fuchs corneal dystrophy in humans without major brain effects [25], while *Ccp5* mutations primarily cause sterility in mice [26]. CCP6 shows a distribution similar to CCP1 but lower cerebellar levels [16], with its highest expression in bone marrow, where its absence impairs megakaryocyte maturation and platelet production [27]. Both CCP4 and CCP6 are functionally homologous to CCP1, shortening glutamate side chains of tubulins [6]. However, CCP4 shows little cerebellar expression or relevance [6,16], whereas CCP6 appears to compensate for CCP1 loss in the cerebral cortex —but not in the cerebellum—based on our findings (see below; Appendix A).

Knockout models have also been developed for CCP2 and CCP3, but their loss is compensated by CCP1, CCP4 and CCP6, resulting in milder phenotypes than those observed with other carboxypeptidases [28]. Several CCPs, except CCP2 and CCP3, show strong testis expression [16], and their loss causes male sterility [26,29]. These findings support the existence of dependency-degeneration patterns, with lobe X representing another case of reduced dependence on *Ccp1* and, consequently, lower vulnerability in PCD mice. Further studies are needed to clarify why lobe X is less reliant on deglutamylation. To explore this, we next analyze *Ccp4*, *Ccp6* and *Ttll1* expression.

CCP4 and CCP6 are functionally homologous to CCP1 and also shorten tubulin glutamate chains [6]. For CCP4, multiple studies have shown low expression in the cerebellum and cerebral cortex, with higher levels in tissues such as the eye [6,16]. Our results agree with these findings: CCP4 protein was undetectable in the cerebellum, and qPCR revealed extremely low *Ccp4* expression (C_T_ values approached 32 vs. 22–27 for other genes; Appendix A). Although these data indicate limited relevance of CCP4 in the cerebellum, small expression changes may still reflect regional differences in deglutamylation dependency. However, this interpretation should be made cautiously given the gene’s minimal expression.

*Ccp4* expression showed minimal variation over time in both cerebellar regions, except at P50, when a slight increase was observed in lobes I to IX. At P20–P25, *Ccp4* expression was lower in lobe X than in the rest of the lobes, suggesting reduced dependence on deglutamylation at these ages. From P30 to P40, differences disappeared, coinciding with the second peak of *Ccp1* expression and its increased requirement for deglutamylation (Figure 8). At P50, *Ccp4* expression again decreased in lobe X while slightly rising in the other lobes, paralleling the decline of *Ccp1* in lobe X at this stage. Overall, *Ccp1* and *Ccp4* expression fluctuated in parallel, though *Ccp4* remained less expressed and functionally less relevant (Appendix A). This is congruent if we consider that genes of the *Ccp* family are paralogous [30] and their functions overlap, which would also suggest they have similar expression patterns.

In PCD mice, *Ccp4* expression at P20 was lower in lobe X than in the rest of the lobes, as in wild-type mice. By P25, however, expression levels equalized across both regions, unlike in the wild-type mice. Although *Ccp4* has little functional relevance in the cerebellum, the slight rise in its expression in lobe X at P25 in PCD mice might reflect a minor compensatory response to *Ccp1* loss. Nevertheless, overall differences between wild-type mice and PCD mice were negligible, confirming that any *Ccp4* changes in PCD animals are minimal.

CCP6, in contrast, appears more relevant than CCP4 in the cerebral cortex and cerebellum (Appendix A) [6]. During early postnatal stages, polyglutamylation is elevated in both wild-type and PCD mice [6]. Later, it normalizes in the cerebral cortex of PCD mice but remains high in the cerebellum [6], suggesting a compensatory role for CCP6 in the cortex but not in the cerebellum. Consistently, *Ccp6* expression was strong in the cortex and lower in the cerebellum [16]. Western blot and qPCR data confirmed comparable expression levels of *Ccp6* and *Ccp1* in the cerebellum (unlike *Ccp4*, see above). Temporally, *Ccp6* expression in wild-type mice decreased at P30 and recovered at P35, paralleling *Ccp1*, while in lobe X it remained stable until P50. A slight increase at P50 in other lobes (Figure 1) further suggests changing requirements for deglutamylation over time.

In parallel, our data supports the hypothesis of a certain compensatory function of CCP6 in the lobe X. At P30—when *Ccp1* expression decreases across the cerebellum—***Ccp6* expression rises specifically in lobe X**, resembling the compensatory mechanism described in the cerebral cortex [14]. Then, there is an increase in the expression of *Ccp6* in lobe X that coincides temporally with the decreased expression of *Ccp1*. This finding has not been described until now and **could explain why this region degenerates later in PCD mice**. The increase in *Ccp6* could make lobe X less dependent on the expression of *Ccp1* at this age. By contrast, at P35 and P40, this difference in expression disappears and both regions exhibit similar expression patterns. Therefore, the compensatory mechanism of *Ccp6* detected at P30 seems to disappear, coinciding with the moment in which lobe X of the PCD mouse begins to degenerate. Finally, at P50, the expression of *Ccp6* in lobe X is less than in the rest of the lobes, similar to *Ccp1* and *Ccp4*. This final age is noteworthy since the temporal expression of *Ccp4* and *Ccp6* increases slightly in lobes I-IX and is lower in lobe X. Therefore, at P50, lobe X appears to be again less dependent on deglutamylation. Future functional experiments should be driven to validate this hypothesis.

In PCD mice, *Ccp6* expression shows no differences between lobe X and the other lobes at any age, paralleling wild-type patterns. However, when comparing genotypes, at P25 lobes I-IX of PCD mice express less *Ccp6* than those of wild-type animals (Figure 4K). This decrease in *Ccp6* expression may be attributed to the neuronal degeneration occurring at this age, specifically the death of Purkinje cells, neurons that normally express this gene [30]. Such ongoing neuronal death may be a limitation to further conclusions affecting a comparative between genotypes in lobes I–IX. Nevertheless, apart from this specific finding at P25, no significant differences in *Ccp6* expression are observed between PCD and wild-type mice in other cerebellar regions or ages. This confirms that the *pcd* mutation does not affect *Ccp6* expression, as has been verified in HEK293T cells carrying this mutation [31].

TTLL1, the enzyme responsible for tubulin glutamylation, plays a key role in neuronal maturation since dynamic microtubules are required in developing neurons, while mature neurons exhibit more stable structures [32]. During early development, polyglutamylation promotes cytoskeletal dynamics, but from P15—when the cerebellar cortex is largely formed—increased *Ccp1* expression in wild-type mice [15] reduces glutamylation and stabilizes microtubules [4]. In PCD mice, this regulation fails, leading to Purkinje cell degeneration [4]. The results from our Western blot analysis confirm the expression of TTLL1 in all tissues and ages studied, including both wild-type and PCD mice. This finding further supports the notion that the hyperglutamylated state of the cytoskeleton in PCD mice does not cancel TTLL1 expression. These data are confirmed by our qPCR analyses (see below).

In wild-type mice, *Ttll1* expression decreases at P30 in all lobes, coinciding with minimal *Ccp1* and *Ccp6* expression. This temporal pattern suggests that reduced TTLL1 lowers glutamylation levels, decreasing the demand for deglutamylation (Appendix A).

Comparing *Ttll1* expression across cerebellar regions, we observed lower expression in lobe X only at P35 and P50. Since *Ccp1* expression is consistently lower in lobe X, but TTLL1 activity appears similar across lobes (except for P35 and P50, a result that will be discussed later on), this suggests that lobe X may remain constitutively more glutamylated. Hyperglutamylation is associated with cytoskeletal immaturity and greater dynamism [4], so a plausible hypothesis is that lobe X tolerates or even requires a more dynamic cytoskeleton due to its simpler and more primitive region [33].

Finally, protein levels of TTLL1 and TTLL7 were slightly lower in PCD mice than in wild-type animals, but without significant differences [31]. Similarly, no differences in *Ttll1* expression were detected between genotypes or cerebellar regions, indicating that the *pcd* mutation does not affect *Ttll1* expression.

Overall, the consistently lower *Ccp1* expression in lobe X compared to other lobes suggests reduced dependency on this deglutamylase, while comparable *Ttll1* expression implies that lobe X can tolerate a more glutamylated, and possibly more flexible, cytoskeleton. These findings align with previous studies reporting no genotype-dependent changes in TTLL1 or CCP expression in PCD models [31].

## 4. Materials and Methods

### 4.1. Mice and Tissue Collection

Wild-type and PCD mice of the C57/DBA hybrid strain were used (45 animals distributed into 8 groups; see Table 1). Due to the guidelines of 3Rs for animal experimentation as well as the limitations for obtaining PCD mice (weak animals, 1/4 of the offspring, see below), each experimental group was limited to 5–7 animals (Table 1). This number avoided a waste of experimental subjects without compromising a good statistical power Animals were established in a colony at the facilities of the Animal Experimentation Service of the University of Salamanca. As PCD males are sterile and females cannot breed with their offspring, PCD mice were obtained by crossing heterozygous animals. To distinguish the three possible resulting genotypes (+/+, +/*pcd* and *pcd*/*pcd*), DNA was extracted from tail samples and PCRs were performed to identify D13Mit250 and D13Mit283 microsatellites. Wild-type and *pcd* alleles have different molecular weights for these markers and can be separated by agarose gel electrophoresis, as previously described [34].

Mice were housed, handled and euthanized in accordance with the stipulations of the Council of the European Communities (2010/63/EU) and the Spanish Legislation (RD118/2021) in force for the use and care of animals. Likewise, the Bioethics Committee of the University of Salamanca approved the procedures carried out), as well as the number of animals (reference numbers 291 and 613).

Mice were anesthetized and sacrificed by cervical dislocation followed by decapitation, and their fresh cerebellum was quickly extracted through the posterior part of the cranium. The brainstem was removed by cutting the cerebellar peduncles to access the cerebellum from its ventral part. Next, lobe X was extracted by inserting a blade through the fissure that separates it from the adjacent lobe IX. Once lobe X was removed, the cerebellar hemispheres were detached from the vermis. This dissection was performed to ensure that the most lateral parts of lobe X were not eliminated while cutting the cerebellar hemispheres.

### 4.2. Quantitative PCR

To analyze the gene expression of *Ccp1*, *Ccp4*, *Ccp6* and *Ttll1*, relative qPCR was performed to facilitate the comparison of gene expression between two tissues. mRNA expression levels were studied using qPCR for two main reasons. Firstly, analyzing mRNA expression would allow the direct comparison of the results with the same genes analyzed at a transcriptional level in previous works [19]. Secondly, for practical reasons, immunohistochemical techniques are not always feasible due to the availability or functionality of specific antibodies against the proteins of interest. By contrast, CCP1, CCP6 and TTLL1 antibodies are compatible with Western blotting, so this technique was used to confirm that the RNA expression of these genes corresponded qualitatively with the corresponding protein levels (see below).

Total RNA was extracted from the tissue of interest using the commercial kit PureLink^TM^ RNA Mini Kit (Invitrogen; Carlsbad, CA, USA). With RNA as a template, copy DNA (cDNA) was reverse-transcribed using the High Capacity cDNA Reverse Transcription Kit (Invitrogen). Finally, cDNA was used for performing qPCR in the thermocycler QuantStudio^TM^ 7 Flex Real-Time PCR System (Applied Biosystems; Foster City, CA, USA) using the oligonucleotides shown in Table 2. We have used triplicates for our samples in all qPCR experiments to reduce their intrinsic variability. The device employed takes into account these replicates and automatically analyzes them to validate each sample. If variability among technical triplicates for a given sample was unacceptably high, the qPCR was repeated. Finally, all qPCR results were validated with the analysis of the corresponding melting curves. Those samples with inadequate melting curves were discarded and the qPCR was repeated.

C_T_ values obtained by qPCR were analyzed by relative quantification (ΔΔC_T_). This analysis gave a fold change value for each sample: the relative amount of target cDNA (corresponding to a gene of interest) normalized using a reference gene (housekeeping) and compared to a control sample. To determine whether gene expression varied over time in the different tissues, the fold change in each cerebellar region (i.e., lobe X or the rest of the vermis) was calculated separately and compared among different ages: P20, P25, P30, P35, P40 and P50. P50 was set as the age of reference for these comparisons, which is when the cerebellum is considered to be stable without remarkable histological, cytochemical or synaptic changes. The statistical comparison of this study was carried out using the Kruskal–Wallis test, with the help of the IBM SPSS Statistics V.26 program (IBM Statistics; Armonk, NY, USA). When significant differences were found, a *post hoc* test was applied to gather the different ages into the most probable groups with statistically similar values.

Additionally, a comparison of the fold changes in the genes of interest between lobe X and the rest of the lobes was done separately at different ages (P20, P25, P30, P35, P40 and P50) using the Mann–Whitney’s *U* test.

Finally, at P20 and P25, the fold changes in genes of interest in lobe X of PCD and wild-type mice were compared. These two specific ages were chosen because at later stages Purkinje cell death in PCD mice is significantly advanced and the analyses would have no longer reflected the gene expression of these cells. Statistical differences were studied using Mann–Whitney’s *U* test.

Non-parametric statistical tests were employed due to (1) the limited number of subjects for each sample, and (2) that not all samples for each variable met the assumption of normality (Kolmogorov–Smirnov test).

### 4.3. Protein Analysis

The qPCR results refer to the expression of the total mRNA of the genes of interest. However, an increase in the amount of mRNA does not necessarily imply an increase in protein expression. Thus, a Western blot analysis was performed to complement and confirm the qPCR results, as this method is both qualitative and semi-quantitative. A smaller number of mice per group was used to avoid using an unnecessary amount of animals, following the directives of animal care. Therefore, a statistical analysis was not carried out. The groups of mice coincided with those used for performing the qPCR analysis: 6 groups of wild-type mice, at P20, P25, P30, P35, P40 and P50, and 2 groups of PCD mice, one at P20 and the other at P25. Two mice from each experimental group were sacrificed and lobe X was dissected using the same procedure carried out on the mice used for qPCR.

The tissue samples were mechanically disrupted with plastic pistils in RIPA buffer (50 mM Tris pH 8, 150 mM NaCl, 1% Igepal, 0.5% sodium deoxycholate and 0.1% sodium dodecyl sulphate, SDS) containing protease inhibitors (Protease Inhibitor Cocktail, Sigma; Saint-Louis, MO, USA). Then, the samples were centrifuged for 10 min at 10,000× *g* at 4 °C. The supernatants were transferred to new tubes where the total protein was quantified using the Bradford assay.

Samples were diluted 1:1 with Laemmli 2X loading buffer (4% SDS, 20% glycerol, 10% β-mercaptoethanol, 0.004% bromophenol blue and 0.125 M Tris-HCl pH 6.8) and incubated at 100 °C for 7 min to denature the proteins and expose their epitopes. Then, the total protein for each sample was loaded onto a 10% polyacrylamide gel. First, the gels were run at 90 V for approximately 45 min in an electrophoresis cuvette with SDS-PAGE buffer. Then, the voltage was increased to 120 V for 1 h to separate the different proteins based on molecular weight.

After the proteins had been separated, they were transferred to a polyvinylidene fluoride membrane activated with 100% methanol for 1 min, by applying a current of 220 mA for 2–2.5 h immersed in transfer buffer. The membrane was then incubated for 1 h under gentle agitation in a solution containing 3% bovine serum albumin (BSA), dissolved in a Tris-buffered saline solution with 0.1% Tween-20 (TBS-T: Tris-HCl 20 mM, NaCl 150 mM and 0.05% Tween^®^-20; Sigma), to avoid non-specific binding.

After blocking with BSA, the membranes were incubated at 4 °C overnight with the primary antiserum against the proteins of interest. The antibody concentrations used, the molecular weights and other data are listed in Table 3 GAPDH was used as the loading control to verify that the technique had been carried out correctly.

The following day the membranes were washed 3 times in TBS-T for 10 min to remove any excess antibody. Then, they were incubated for 50 min with a secondary antibody coupled to horseradish peroxidase (HRP) at 1:10,000 in TBS-T with 5% skimmed milk powder. From this step onwards, the entire process was carried out in the dark to avoid HRP activation. Finally, the membranes were again washed 3 times with TBS-T for 10 min.

The membranes were developed using a chemiluminescent detection kit (Advansta; San Jose, CA, USA) and the reaction was detected by a MicroChemi 4.2 device (DNR Bio-Imaging Systems, Jerusalem, Israel). This system allowed us to take several images of the developed membranes at different exposures, enabling more sensitive detection of low-abundance proteins.

Protein expression was determined by densitometric analysis using ImageJ software (V.1.54f; Wayne Rasband, National Institutes of Health; Bethesda, MD, USA). Total integrated density was obtained for each band (CCP1, CCP6, TTLL1 and GAPDH) and background was subtracted to obtain the specific integrated density for each protein. Relative expression was calculated as the relationship CCP1/GAPDH, CCP6/GAPDH or TTLL1/GAPDH for each sample. The results obtained for CCP1 were analyzed using a three-way ANOVA test with Graph Pad Prism software (V.10.6.1; GraphPad Software; Boston, MA, USA).

## 5. Conclusions

In summary, the lower vulnerability of lobe X in PCD mice could be due to several factors:Lower expression of *Ccp1* in lobe X compared to other lobes.Constitutive hyperglutamylation leads to a more dynamic and less stable cytoskeleton.The simpler and more primitive nature of lobe X may inherently require or tolerate greater cytoskeletal instability.Additionally, increased resistance characterizes lobe X ([21]; also see Hernández-Pérez et al., submitted to this same journal): it shows a particularly increased basal expression of HSP25, known for its protective properties, and the neuronal loss of PCD mice triggers both its presence and its active form. This constitutive protection against any type of neural damage is additional to its resistance to the effects of the lack of CCP1.

And key question is *why is lobe X different from other lobes?* Lobe X of the cerebellum is more resistant than the other lobes in numerous animal models suffering Purkinje cell death, and it is mainly due to an increased expression of HSP25 (for a review see [21]). Additionally, in the present work we have detected a lower vulnerability in PCD mice, probably not exclusive to this model of cerebellar degeneration. Why do both factors (higher protection and lower vulnerability) reside in the same lobe? Why might lobe X be more important than the rest of the cerebellar cortex? Lobe X belongs to the flocculonodular zone, the most ancient cerebellar region, which plays a key role in the most basic functions of the cerebellum [33]. Thus, the protection of the flocculonodular zone might be a priority, more than that of other cerebellar areas.

Lobe X has unique characteristics that make it less vulnerable to degeneration in PCD mice, probably due to a combination of the aforementioned factors. Once identified, these characteristics could be useful to develop strategies to prevent Purkinje cell loss in CONDCA patients and may help alleviate the symptoms of other neurodegenerative diseases.

## Figures and Tables

**Figure 1 ijms-26-10378-f001:**
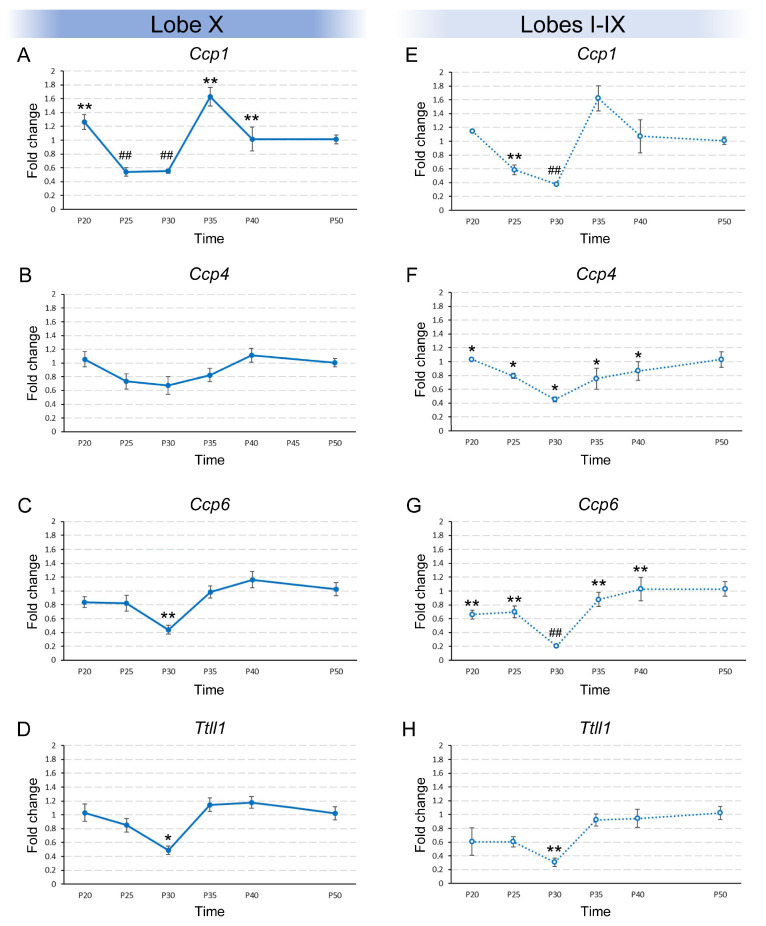
Temporal expression of *Ccp1*, *Ccp4*, *Ccp6*, and *Ttll1* in vermis of wild-type animals from P20 to P50. Results are separated in lobe X ((**A**–**D**), continuous lines) and lobes I-IX ((**E**–**H**), dashed lines). (**A**) Significant differences in *Ccp1* expression in lobe X were observed over time. (**B**) No significant differences were observed over time for *Ccp4* and extremely low expression levels were detected (C_T_ around 32). (**C**,**D**) The same expression pattern was observed in lobe X for both *Ccp6* and *Ttll1*, with significant differences. (**E**) *Ccp1* expression in lobes I-IX decreased progressively, and then rose back to initial levels, remaining stable. (**F**) *Ccp4* expression remained constant from P20 to P40, and at P50 showed a distinct expression level. (**G**) *Ccp6* expression slightly decreased at P30 and then returned to initial levels at P35 and P40, and increased again at P50. (**H**) *Ttll1* expression decreased at P30 and then rose back to initial levels. Data are presented as mean fold change ± standard error of the mean. When Kruskal–Wallis indicated significance, step-down post hoc comparisons defined the most likely subsets; P50 was considered the reference level of gene expression; equal symbols indicate statistically similar groups; * *p* < 0.05; ** or ## *p* < 0.01.

**Figure 2 ijms-26-10378-f002:**
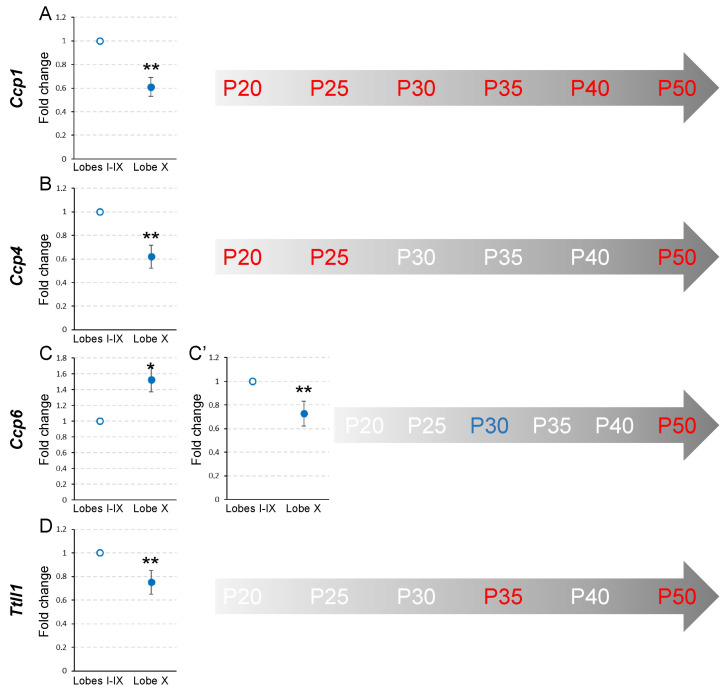
Comparison of *Ccp1*, *Ccp4*, *Ccp6*, and *Ttll1* expression between lobe X and the rest of the vermis in wild-type mice from P20 to P50. Charts show a representative example of the fold change in each gene at different ages. In the timelines (gray arrows) those ages at which statistical differences were found are colored in red (a decrease in gene expression for lobe X in comparison with the rest of the vermis) or blue (an increase in gene expression for lobe X). (**A**) The expression of *Ccp1* was lower in lobe X compared to the other lobes, from P20 to P50. (**B**) *Ccp4* presented a lower expression in lobe X compared to the other lobes at P20, P25 and P50. (**C**,**C’**) The expression of *Ccp6* fluctuated more: at P30 (**C**), it was higher in lobe X than in the other lobes; conversely, at P50 (**C’**) lobe X presented a lower gene transcription. (**D**) *Ttll1* presented lower gene expression in lobe X at P35 and P50. Note that the expression of *Ccp1* was always reduced in lobe X, and that at P50 all the analyzed genes presented a lower production in lobe X. Data are presented as mean fold change ± standard error of the mean. * *p* < 0.05; ** *p* < 0.01.

**Figure 3 ijms-26-10378-f003:**
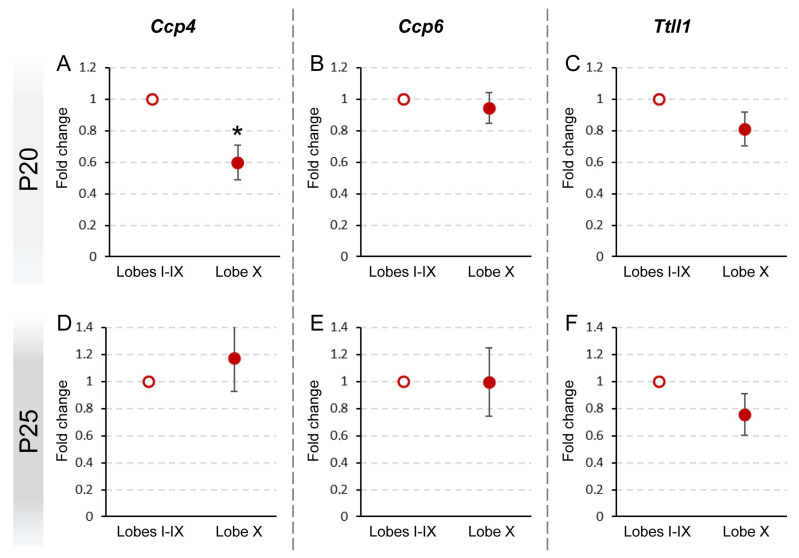
Comparison of *Ccp4*, *Ccp6*, and *Ttll1* expression between lobe X and the rest of the vermis in PCD at P20 and P25. Significant differences were found only in the expression of *Ccp4* at P20 (**A**), which appeared reduced in lobe X. The other genes did not show differences either at P20 (**B**,**C**) or P25 (**D**–**F**). Data are presented as mean fold change ± standard error of the mean. * *p* < 0.05.

**Figure 4 ijms-26-10378-f004:**
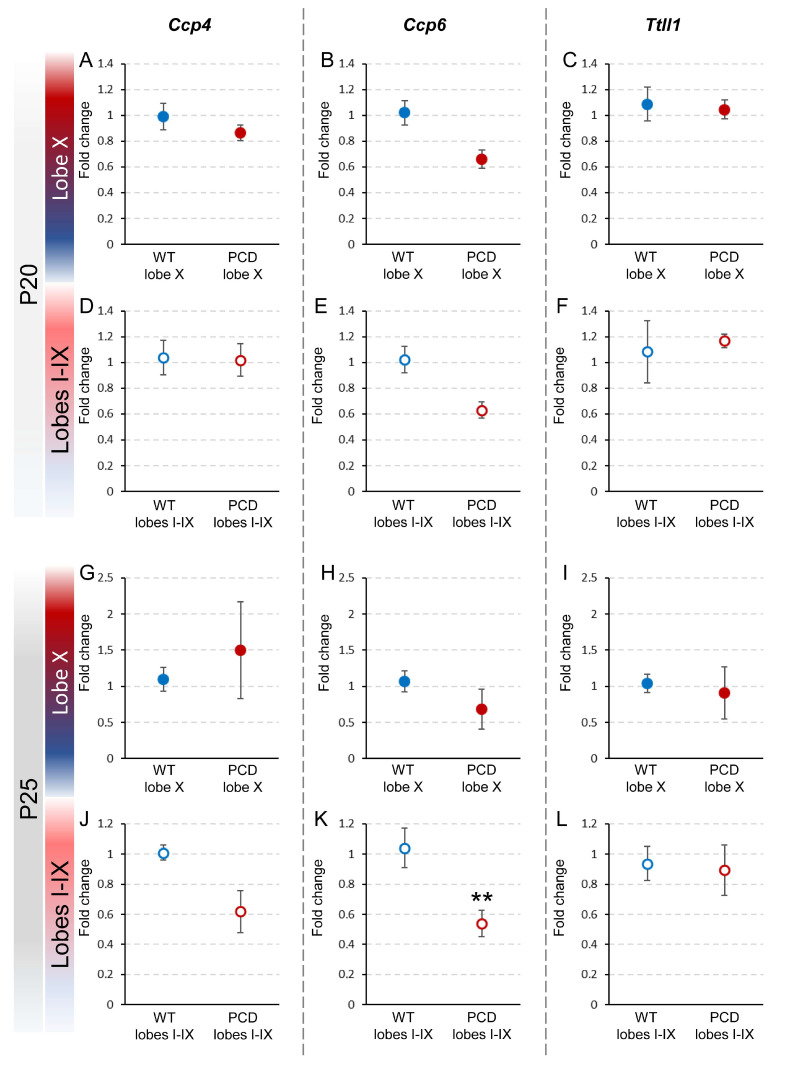
Relative expression of *Ccp4*, *Ccp6*, and *Ttll1* between wild-type and PCD mice at P20 and P25. The upper graphs (**A**–**F**) show the comparison of the results obtained at P20 between wild-type and PCD animals, either for lobe X (**A**–**C**) or the rest of the vermis (**D**–**F**). The lower graphs (**G**–**L**) show the comparison of the results obtained at P25 between genotypes, either for lobe X (**G**–**I**) or the rest of the vermis (**J**–**L**). Differences were found only for *Ccp6* in lobes I–IX at P25 (**E**), with lower expression in PCD mice. Data are presented as mean fold change ± standard error of the mean. WT, wild type. ** *p* < 0.01.

**Figure 5 ijms-26-10378-f005:**
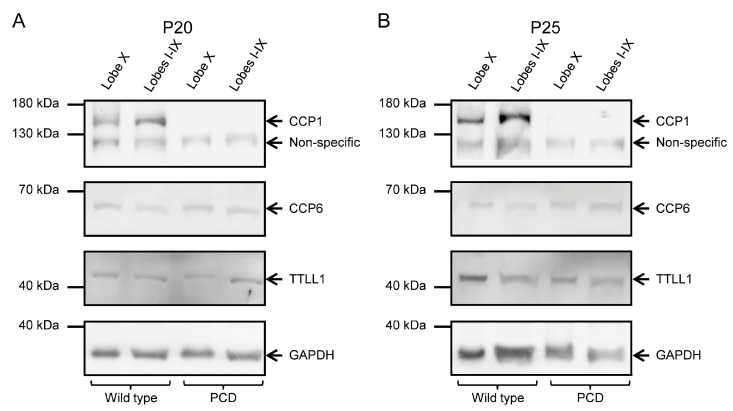
CCP1, CCP6, and TTLL1 protein expression at P20 and P25 in wild-type and PCD animals. (**A**) The left images show the results corresponding to P20. In the top box, CCP1 is present in wild-type mice but absent in PCD animals; the lower band below CCP1 corresponds to non-specific binding, as indicated by the antibody manufacturer. Additionally, the presence of CCP6 and TTLL1 was confirmed in all tissues and genotypes. (**B**) The right pictures display the results at P25. Again, CCP1 expression is absent in PCD mice; in the control vermis, the CCP1 band appears more intense than in lobe X. Once again, the presence of CCP6 and TTLL1 was confirmed in all tissues and genotypes. The lower band in both image series shows GAPDH, used as a loading control.

**Figure 6 ijms-26-10378-f006:**
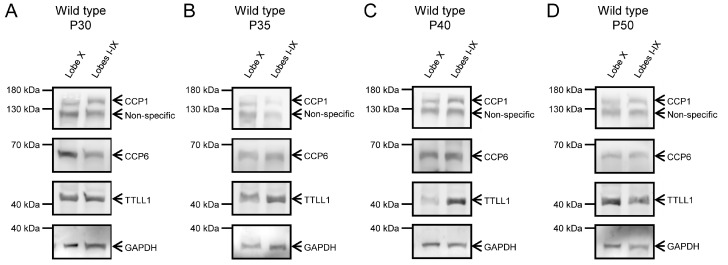
CCP1, CCP6, and TTLL1 protein expression at P30, P35, P40, and P50 in wild-type animals. Protein expression of CCP1, CCP6, and TTLL1 was confirmed in wild-type mice at P30 (**A**), P35 (**B**), P40 (**C**), and P50 (**D**) across all regions and ages. At P40 (**C**), TTLL1 expression appears qualitatively reduced in lobe X. The lower band shows GAPDH, used as a loading control in both image series.

**Figure 7 ijms-26-10378-f007:**
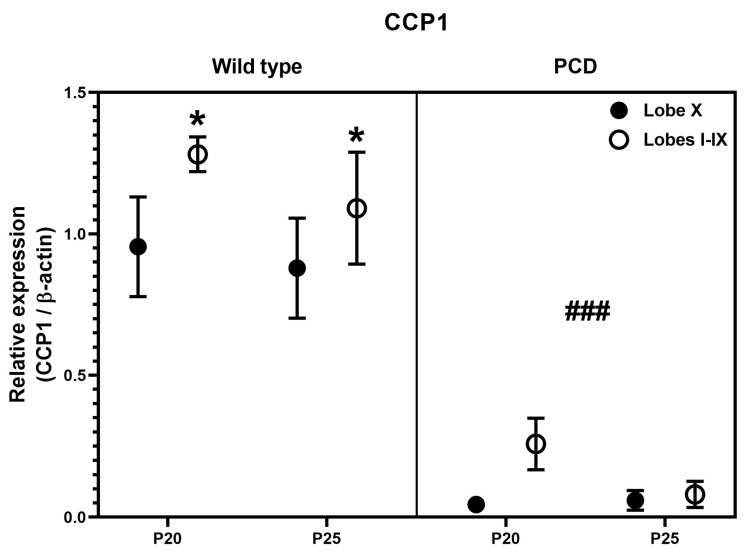
Quantification of CCP1 expression after Western blot analyses in wild-type and PCD mice at P20 and P25, comparing lobe X and the rest of cerebellum. Statistically significant differences were detected for both genotype and cerebellar region. * *p* < 0.01 for region, ### *p* < 0.01 for genotype, general differences after three-way ANOVA analysis.

**Figure 8 ijms-26-10378-f008:**
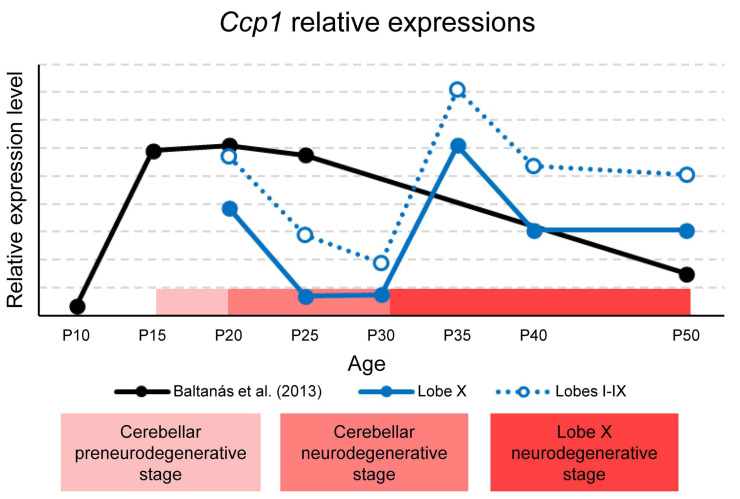
Representation of *Ccp1* expression in the whole cerebellum, lobe X, and the rest of lobes in wild-type mice. Data for lobe X (blue line) and lobes I-IX (blue dotted line) are gathered from the present work. Data represented in black were obtained from our same laboratory and published in [19]. The horizontal red band indicates—in three shades of red—the preneurodegenerative and neurodegenerative stages of the cerebellum and the hypothetical delayed neurodegenerative stage of lobe X. Although graphs cannot be compared mathematically, the results for lobe X are shown below those for the rest of the lobes to refer to the lower expression obtained when comparing both regions at each age considered. It should be noted that Baltanás et al. [19] did not include intermediate time points between P25 and P50 and analyzed the entire cerebellum, whereas the present study assessed specific regions. These methodological differences likely explain the transient P35–P40 increase observed in our data but not in the previous study.

**Table 1 ijms-26-10378-t001:** Number of animals distributed by age and genotype. Note that no PCD mice older than P30 were used because their Purkinje cells have disappeared.

Genotype	P20	P25	P30	P35	P40	P50
Wild type	5	6	5	5	6	6
PCD	5	7	-	-	-	-

**Table 2 ijms-26-10378-t002:** Oligonucleotides used for qPCR analysis.

Gene Targeted	Sequence	Oligonucleotide	Source
*Ccp1*	Sense	CCCCATTGTAGTTCCCACAG	[19]
	Antisense	CTTCCTTGGCTTCCTCTCCT	
*Gapdh*	Sense	GCCTATGTGGCCTCCAAGGA	
	Antisense	GTGTTGGGTGCCCCTAGTTG	
*Ccp4*	Sense	GATCGCAAACTGGGAGTACAG	Roche’s Universal ProbeLibrary Assay Design Center online Tool
	Antisense	TGGGCTCGTCTAAACTGAAGA	
*Ccp6*	Sense	TCTTCATTTGTGTGCCAAGG	
	Antisense	GACTAAATGTTCTCGTAGGACACG	
*Ttll1*	Sense	CCCAAGGAAGTACTTGGCAAC	
	Antisense	GGCGGTTTCTAAGCTCTCG	

**Table 3 ijms-26-10378-t003:** Western Blot antibodies.

Type of Antibody	Molecule Labeled ^1^	Species	Dilution or Concentration	Antigen Weight	Producer
Primary	CCP1	Rabbit	1:2000	137 kDa	Proteintech (Rosemont, Shakopee, MN, USA)
Primary	CCP6	Rabbit	1:1000	62 kDa	MyBioSource (San Diego, CA, USA)
Primary	TTLL1	Rabbit	1:1000	49 kDa	ThermoFisher (Madrid, Spain)
Primary	GAPDH	Mouse	1 μg/μL	35 kDa	Applied Biosystems Foster City, CA, USA
Secondary	HRP ^2^ anti-Rabbit	Goat	1:10,000	-	Jackson ImmunoResearch (West Grove, PA, USA)
Secondary	HRP anti-Mouse	Goat	1:10,000	-	Jackson ImmunoResearch (West Grove, PA, USA)

^1^. The reporter molecule of secondary antibodies is also indicated. ^2^. HRP: horseradish peroxidase.

## Data Availability

The raw data supporting the conclusions of this article will be made available by the authors on request.

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
