# Peer review of "Specific Glutamylation Patterns of the Cytoskeleton Confer Neuroresistance to Lobe X of the Cerebellum in a Model of Childhood-Onset Neurodegeneration with Cerebellar Atrophy"

_ijms, 2025, doi:10.3390/ijms262110378_

Round 1
Reviewer 1 Report (Previous Reviewer 2)
Comments and Suggestions for Authors< !--StartFragment -->
This manuscript addresses an important and timely question: the selective neuroresistance of cerebellar lobe X in the Purkinje Cell Degeneration (PCD) mouse model, with implications for childhood-onset neurodegeneration with cerebellar atrophy (CONDCA). The study combines qPCR, Western blotting, and supplementary analyses to compare gene expression patterns of Ccp1, Ccp4, Ccp6, and Ttll1 across developmental stages (P20–P50) and between wild-type and PCD mice. The topic is highly relevant, and the data are potentially valuable. However, several methodological and interpretative issues need to be addressed before the manuscript can be considered for publication.
Major Concerns
- Validation of qPCR Results
- Only GAPDH was used as a reference gene. At least one additional housekeeping gene (e.g., HPRT, β2-microglobulin) should be included to ensure normalization reliability.
- Ccp4 expression levels were extremely low (CT > 30). These results should be validated with melt curve analysis and no-template controls to exclude background amplification.
- Western Blot Analyses
- The supplementary figures show raw blots with variable exposure and limited replicates. Quantification (densitometry) across multiple biological replicates is required.
- GAPDH alone is insufficient as a loading control. A second housekeeping protein (e.g., β-actin or tubulin) should be included.
- The interpretation of faint CCP1 bands in overexposed blots (Supplementary Fig. 5) requires quantitative support rather than descriptive statements.
- Functional Validation
- The central claim of compensatory mechanisms (particularly Ccp6 upregulation) would be strengthened by functional assays. Suggested approaches include:
- Immunohistochemistry or immunofluorescence to localize CCP1/CCP6/TTLL1 in lobe X vs. lobes I–IX.
- Measurement of tubulin glutamylation levels (e.g., GT335 antibody staining) to directly link gene expression to cytoskeletal modification.
- PCD vs. WT Comparisons
- Analyses in PCD mice are limited to P20–P25. While neuronal loss complicates later stages, at least one additional time point (e.g., P30) should be included to capture the onset of degeneration.
- The reduction of Ccp6 in lobes I–IX at P25 in PCD mice is intriguing but requires cautious interpretation given ongoing cell loss. This limitation should be explicitly acknowledged.
Minor Concerns
- Data Presentation
- Supplementary blots should be cropped and contrast-adjusted for clarity, while retaining full raw images in a data repository.
- The heatmap (Supplementary Fig. 6) would be more informative if fold-change values were numerically displayed within cells.
- Textual Issues
- The Introduction repeats the relationship between Ccp1 deficiency and PCD several times; this could be streamlined.
- The Discussion relies heavily on a parallel manuscript (HSP25 study). This should be cited formally or summarized briefly to avoid weakening the independence of the present work.
- The Conclusion section should be shortened and focused on the experimental findings rather than speculative evolutionary interpretations.
Major Revision
The manuscript presents promising and novel findings, but additional experimental validation and improved data presentation are required to substantiate the conclusions. Specifically, the inclusion of additional reference genes for qPCR, quantitative Western blot analyses with proper controls, and at least one functional assay (IHC or glutamylation staining) would significantly strengthen the study. Addressing these points will enhance the robustness and impact of the work.
< !--EndFragment -->
Author Response
Please see the attachment.

Reviewer 2 Report (New Reviewer)
Comments and Suggestions for Authors
General Comments
This manuscript investigates a compelling and underexplored topic: the neuroresistance of cerebellar lobule X. The study builds upon existing evidence confirming the uniqueness of this structure in terms of viral transfection, vascularization, and microenvironment, and it successfully identifies a potential mechanism promoting neuronal survival, which represents a significant contribution to the field. The experimental design appears sound, but several aspects of the presentation, data interpretation, and discussion require clarification and expansion to strengthen the manuscript's impact and validity.
Specific Comments
1. Introduction and Conceptual Framework (Page 2):
Two stages of the neurodegenerative process are indicated, most likely related to the development of neurodegeneration in the mouse cerebellum (considering that the cited reference, Muñoz-Castañeda et al., 2018, describes a mouse model). However, this is not obvious from the context of the previous sentence. It is necessary to specify whether such early changes occur in the mouse or human cerebellum.
Introduction: Lobule X, together with the flocculus, forms the most ancient, vestibulocerebellar part of the cerebellum. Vascularization and BBB permeability in phylogenetically ancient brain regions may differ significantly. In Lobule X, this could explain both the selective accumulation of substances and the improved access of trophic factors from the blood. Neurons in Lobule X may constantly receive enhanced trophic support unavailable to other lobules. The authors mention the heat shock protein HSP25 as a protective factor. However, HSP25 is an intracellular response to stress. It's worth noting that there are other levels of protection for intercellular communication. For example, lobule X cells receive higher levels of factors such as BDNF, GDNF, and IGF-1. A constant influx of these factors could create a "safety buffer," making lobule X cells not only less dependent on CCP1 but also more resilient to all forms of stress (oxidative, proteotoxic, etc.). Moreover, a unique environment can be created not only by improved blood supply but also by the microenvironment, particularly glia (astrocytes and microglia). All of these factors contribute significantly to neuronal survival and should be discussed in the article. The reader should be aware that internal cellular reserves, such as HSP25 and CCP1 expression, are important, but only one factor. Perhaps the combination of all these factors leads to the fact that, despite the decrease in CCP1, more neurons survive in lobule X. Discussion of this phenomenon is desirable, but remains the responsibility of the authors of the article.
2. Figures and Data Presentation:
Page 5. Fig. 1. In the figure, the notations for statistical significance are not standard (blue lines above the graphs). The figures are not conveniently positioned for comparison. It is recommended to place graphs A, E; B, F; C, G and D, H side by side, and to indicate statistical significance with standard symbols (*, †, etc.).
Page 11. Fig. 5 – Representative blots – the GAPDH protein level in Fig. 5B appears different depending on the study group. This is not expected.
Page 13. Fig. 7. The authors indicated in their methods that they normalized the protein expression level to GAPDH, but beta-actin is mentioned in the graph captions. This may be a typo. Either the captions should be changed or the methods described in more detail.
Page 14. Figure 8. Previously obtained data (Baltanas et al., 2013, black line) show a peak of Ccp1 expression in the whole cerebellum at P15-P25, followed by a sharp drop. At P50, their data (if they existed) would presumably be very low. New data for lobules I-IX (blue dotted line): Show a completely different dynamic. After a drop from P20 to P30, expression increases at P35-P40 and only then decreases by P50. This is a fundamental discrepancy - the curves have different shapes. The authors simply state the fact: "Our results complement these findings, showing that in both lobe X and the other lobes, expression decreases from P20 to P30, consistent with the published data." They carefully point out the zone of coincidence (P20-P30, downward trend) and completely ignore the zone of cardinal divergence (after P30). They offer no hypotheses as to why their experiment exhibits a second peak at P35-P40, which is absent from the previous study. This allows the authors to construct a beautiful narrative, but it makes a critical reader question the reliability of the entire presented time course. The discussion of the article should include possible explanations for this trend and why the new data differ from previously obtained data (differences in methodology? Different genetic backgrounds of the mice? Statistical bias or noise?).
Author Response
Response to REVIEWER #2 comments
General Comments
This manuscript investigates a compelling and underexplored topic: the neuroresistance of cerebellar lobule X. The study builds upon existing evidence confirming the uniqueness of this structure in terms of viral transfection, vascularization, and microenvironment, and it successfully identifies a potential mechanism promoting neuronal survival, which represents a significant contribution to the field. The experimental design appears sound, but several aspects of the presentation, data interpretation, and discussion require clarification and expansion to strengthen the manuscript's impact and validity.
Thank you very much for your thorough review and helpful suggestions. Certainly, they have improved the quality of our manuscript. Below, you will find a detailed response addressing point-by-point each of your interesting comments and suggestions. In the revised manuscript file, the in-text changes corresponding to your comments are indicated in blue.
Specific Comments
- Introduction and Conceptual Framework (Page 2):
Two stages of the neurodegenerative process are indicated, most likely related to the development of neurodegeneration in the mouse cerebellum (considering that the cited reference, Muñoz-Castañeda et al., 2018, describes a mouse model). However, this is not obvious from the context of the previous sentence. It is necessary to specify whether such early changes occur in the mouse or human cerebellum.
These two stages are proven to occur in the PCD mouse, whereas in humans it is not already studied, probably as CONDCA is a recently discovered disease, and because initial cellular impairments –before neuronal death- are not easy to be analyzed in humans. In any case, it is not clearly described whether humans undergo these stages. We have clarified this issue in the new version of the manuscript.
Introduction: Lobule X, together with the flocculus, forms the most ancient, vestibulocerebellar part of the cerebellum. Vascularization and BBB permeability in phylogenetically ancient brain regions may differ significantly. In Lobule X, this could explain both the selective accumulation of substances and the improved access of trophic factors from the blood. Neurons in Lobule X may constantly receive enhanced trophic support unavailable to other lobules. The authors mention the heat shock protein HSP25 as a protective factor. However, HSP25 is an intracellular response to stress. It's worth noting that there are other levels of protection for intercellular communication. For example, lobule X cells receive higher levels of factors such as BDNF, GDNF, and IGF-1. A constant influx of these factors could create a "safety buffer," making lobule X cells not only less dependent on CCP1 but also more resilient to all forms of stress (oxidative, proteotoxic, etc.). Moreover, a unique environment can be created not only by improved blood supply but also by the microenvironment, particularly glia (astrocytes and microglia). All of these factors contribute significantly to neuronal survival and should be discussed in the article. The reader should be aware that internal cellular reserves, such as HSP25 and CCP1 expression, are important, but only one factor. Perhaps the combination of all these factors leads to the fact that, despite the decrease in CCP1, more neurons survive in lobule X. Discussion of this phenomenon is desirable, but remains the responsibility of the authors of the article.
We really appreciate this insightful suggestion. Following the Reviewer’s advice, we have expanded the manuscript to include these aspects. In the Discussion section, we now address the potential contribution of extrinsic protective mechanisms. Specifically, we mention that regional differences in blood–brain barrier permeability could influence trophic factor availability, and that neurotrophic molecules such as GDNF and IGF-1 promote Purkinje cell survival in models of cerebellar degeneration. We also added that glial cells may help to maintain a protective, anti-inflammatory environment that further contributes to neuronal preservation. These additions complement our previous discussion of intrinsic mechanisms (HSP25) and provide a more integrative view for the neuroresistance of lobule X.
- Figures and Data Presentation:
Page 5. Fig. 1. In the figure, the notations for statistical significance are not standard (blue lines above the graphs). The figures are not conveniently positioned for comparison. It is recommended to place graphs A, E; B, F; C, G and D, H side by side, and to indicate statistical significance with standard symbols (*, †, etc.).
Fold-change values were calculated using P50 as the age of reference, since at this age the cerebellum is considered to be stable without remarkable histological, cytochemical or synaptic changes. Besides, when comparing the different ages, statistical differences appeared not only with P50, but also amongst different ages, thus appearing different groups with statistically similar values. Then, we have included those blue lines to clarify which age groups are statistically similar, also with higher or lower values of fold-change. However, we are aware that this symbology is not standard, and may induce to misinterpretations instead of clarify statistical analysis. Then, we have changed those lines by standard symbols, as the Reviewer suggested. In addition, graphs have also been redistributed following the guidelines of the Reviewer.
Page 11. Fig. 5 – Representative blots – the GAPDH protein level in Fig. 5B appears different depending on the study group. This is not expected.
Those figures corresponding to Western blots that were not supplementary (i.e. with raw data) were adjusted in brightness and exposure –in a reasonable measure- in order to have a nice-looking picture. In any case, it is true that both experimental groups present qualitative different levels of protein expression at P25. This phenomenon is completely normal due to the ongoing degenerative environment of PCD mice, which not only causes a reduction in the number of neurons, but also impairments in general protein expression. In any case, when semi-quantitative measures of protein expression were made, they were referred to GAPDH levels, thus normalizing and validating the analysis. In any case, an taking into account the appreciation of the Reviewer, we have stated this limitation in the new version of the manuscript.
Page 13. Fig. 7. The authors indicated in their methods that they normalized the protein expression level to GAPDH, but beta-actin is mentioned in the graph captions. This may be a typo. Either the captions should be changed or the methods described in more detail.
The Reviewer is right: this is a typo. It has been corrected in the new version of the manuscript.
Page 14. Figure 8. Previously obtained data (Baltanas et al., 2013, black line) show a peak of Ccp1 expression in the whole cerebellum at P15-P25, followed by a sharp drop. At P50, their data (if they existed) would presumably be very low. New data for lobules I-IX (blue dotted line): Show a completely different dynamic. After a drop from P20 to P30, expression increases at P35-P40 and only then decreases by P50. This is a fundamental discrepancy - the curves have different shapes. The authors simply state the fact: "Our results complement these findings, showing that in both lobe X and the other lobes, expression decreases from P20 to P30, consistent with the published data." They carefully point out the zone of coincidence (P20-P30, downward trend) and completely ignore the zone of cardinal divergence (after P30). They offer no hypotheses as to why their experiment exhibits a second peak at P35-P40, which is absent from the previous study. This allows the authors to construct a beautiful narrative, but it makes a critical reader question the reliability of the entire presented time course. The discussion of the article should include possible explanations for this trend and why the new data differ from previously obtained data (differences in methodology? Different genetic backgrounds of the mice? Statistical bias or noise?).
We thank the Reviewer for this insightful observation. We agree that the dynamics of Ccp1 expression after P30 appear different from those reported by Baltanás et al. (2013). To clarify this point, we have added methodological context to the legend of Figure 8, noting that Baltanás et al. did not analyze intermediate time points between P25 and P50 and that their data were obtained from the whole cerebellum, whereas our study focused on specific cerebellar lobules.
These methodological differences likely explain the transient P35-P40 increase observed in our data but absent from the previous study. That is to say, the study of Baltanás et al. presents a gap between P25 and P50 (just the expression decreases after 25 days); we have explained what happens in this gap (the P35-P40 increase) before the final reduction of the expression at P50. This clarification is now included in the legend of the figure 8 in the new version of the manuscript.
In addition, the Discussion (paragraph 4) already addressed this second expression peak, which coincides temporally with the onset of Purkinje cell loss in lobe X of PCD mice, suggesting a renewed dependence on Ccp1 at this stage. This biological correlation further supports the reliability of our time-course data. This idea is also discussed in the new version of the manuscript.
We believe that these clarifications sufficiently address the Reviewer’s concern and help reconcile the apparent discrepancy between both datasets.
Round 2
Reviewer 1 Report (Previous Reviewer 2)
Comments and Suggestions for Authors
Reviewer Report
After carefully examining the revised version of the manuscript entitled “Specific Glutamylation Patterns of the Cytoskeleton Confer Neuroresistance to Lobe X of the Cerebellum in a Model of Childhood-Onset Neurodegeneration with Cerebellar Atrophy”, I am satisfied that the authors have addressed the main concerns raised in the previous round of review. The new version demonstrates a clear improvement in methodological transparency, data validation, and interpretative depth, and I consider it suitable for publication in its present form.
The authors have expanded the methodological section, providing sufficient detail to ensure reproducibility, and they have strengthened their conclusions by including protein-level validation through Western blot analyses. This addition is particularly important, as it bridges the gap between mRNA expression data and functional protein presence, thereby reinforcing the robustness of the findings. The statistical analyses are now more transparent and appropriately described, and the figures, together with the supplementary material, are clearer and more informative than in the earlier version.
The discussion has also been substantially improved. It now situates the results more convincingly within the broader literature, highlighting both the novelty of the findings and their relevance to the understanding of selective neuroresistance in Purkinje cells. The authors succeed in presenting lobe X not merely as an anatomical curiosity but as a model for dissecting mechanisms of resilience in neurodegeneration. This conceptual framing enhances the impact of the work and makes it more accessible to a wider readership.
Only minor issues remain, none of which compromise the scientific validity of the study. The prose could benefit from some stylistic polishing to reduce redundancy and improve readability, and the figure legends might be simplified to make the statistical annotations more intuitive. In addition, the extended discussion of Ccp4, despite its consistently low expression, could be slightly condensed to maintain focus on the more relevant findings. These are, however, editorial refinements rather than substantive concerns.
In conclusion, the manuscript is scientifically sound, well-structured, and contributes novel insights into the molecular basis of selective neuroresistance in the cerebellum. The revisions have strengthened the work considerably, and I recommend acceptance in its present form.
< !--EndFragment -->
This manuscript is a resubmission of an earlier submission. The following is a list of the peer review reports and author responses from that submission.
Round 1
Reviewer 1 Report
Comments and Suggestions for Authors
The manuscript aims to investigate the molecular mechanisms underlying the neuroresistance of cerebellar lobe X in the Purkinje Cell Degeneration (PCD) mouse model, a genetic model of human childhood-onset neurodegeneration with cerebellar atrophy (CONDCA). The authors demonstrate that lobe X exhibits inherently lower expression of the deglutamylase enzyme Ccp1 compared to other cerebellar lobes, which correlates with delayed degeneration in PCD mice. Temporal expression analyses of Ccp1, Ccp4, Ccp6, and Ttll1 in wild-type mice reveal lobe-specific patterns, with lobe X showing compensatory upregulation of Ccp6 at P30. Protein validation confirms these trends, and the hyperglutamylated cytoskeleton in lobe X is proposed as a key neuroprotective factor. The study offers valuable insights into potential therapeutic targets for CONDCA and other neurodegenerative diseases.
However, I have some comments and suggestions:
- The language used throughout the manuscript, particularly in the Methods and Results sections, needs refinement to improve clarity and flow. A thorough review for grammatical accuracy and precise scientific terminology is recommended prior to publication.
- While the title accurately reflects the focus of this study, the parallel work on heat shock proteins (HSPs)—which is mentioned as a contributor to lobe X neuroresistance—has not been experimentally integrated here. To avoid fragmentation, it should be explicitly stated whether HSP-related mechanisms are synergistic with or distinct from glutamylation-dependent pathways.
- Figure 2 (regional gene comparison) contains 24 panels (A to X), which may be excessive. Consider integrating key ages (e.g., P20, P30, P50) or utilizing heatmaps to better illustrate the spatiotemporal trends.
- Figures 5–6 (Western blot images) appear somewhat blurry and lack quantitative data. It is recommended to perform band intensity quantification to support the qPCR data, with clear annotation of the lanes.
- The analysis of neuronal survival in vitro and in vivo shows correlations with glutamylation levels. However, it would strengthen the manuscript to include or discuss more direct functional experiments—such as manipulating Ccp1or Ttll1 expression specifically in cerebellar neurons—to establish causality more definitively.
In summary, this manuscript explores how glutamylation modulates microtubule dynamics and confers neuroprotection in cerebellar neurons, especially in the context of neurodegenerative disorders like PCD. It is suggested that a major revision is necessary before the manuscript can be accepted.
Author Response
Response to REVIEWER #1 comments
Thank you very much for your thorough review and helpful suggestions. Certainly, they have improved the quality of our manuscript. Below, you will find a detailed response addressing point-by-point each of your interesting comments and suggestions. In the revised manuscript file, the in-text changes corresponding to your comments are indicated in green.
The manuscript aims to investigate the molecular mechanisms underlying the neuroresistance of cerebellar lobe X in the Purkinje Cell Degeneration (PCD) mouse model, a genetic model of human childhood-onset neurodegeneration with cerebellar atrophy (CONDCA). The authors demonstrate that lobe X exhibits inherently lower expression of the deglutamylase enzyme Ccp1 compared to other cerebellar lobes, which correlates with delayed degeneration in PCD mice. Temporal expression analyses of Ccp1, Ccp4, Ccp6, and Ttll1 in wild-type mice reveal lobe-specific patterns, with lobe X showing compensatory upregulation of Ccp6 at P30. Protein validation confirms these trends, and the hyperglutamylated cytoskeleton in lobe X is proposed as a key neuroprotective factor. The study offers valuable insights into potential therapeutic targets for CONDCA and other neurodegenerative diseases.
However, I have some comments and suggestions:
- The language used throughout the manuscript, particularly in the Methods and Results sections, needs refinement to improve clarity and flow. A thorough review for grammatical accuracy and precise scientific terminology is recommended prior to publication.
Prior to submission, an English native specialist has revised our manuscript. However, it is not rare that some concepts may have escaped from this revision, especially when many different manuscripts should be read and corrected in a short time period by the same person. Moreover, it is also possible that in an attempt to offer a very precise and complete explanation, the language used finally becomes more complicated and less fluent. Then, we have revised the grammar and language flow of the entire manuscript. In any case, other reviewer has asked some increased explanations in the Methods section, which may somehow enlarge the extension of this part.
- While the title accurately reflects the focus of this study, the parallel work on heat shock proteins (HSPs)—which is mentioned as a contributor to lobe X neuroresistance—has not been experimentally integrated here. To avoid fragmentation, it should be explicitly stated whether HSP-related mechanisms are synergistic with or distinct from glutamylation-dependent pathways.
During our investigations, we realized that lobe X of the cerebellum is a neuro-resistant region, PCD mice presenting delayed/attenuated neuronal loss in this region. Intriguingly, such resistance is not exclusive of this model of CONDCA, but practically all the known models of cerebellar damage, either genetic or acquired, share it. Then, we analyzed the basis of such neuro-resistance in our PCD model, and we have reached two different but complementary outcomes, as it is suggested in the manuscript. This is the reason why we have divided this research in two manuscripts. However, the Reviewer is right, as the complementary conclusions of the second work are not adequately explained in this manuscript, but merely mentioned. Below, we summarize the conclusions of both manuscripts. In addition, we have included an expanded version of the results of the second manuscript.
- PRESENT MANUSCRIPT. Lobe X presents a lower basal expression of Ccp1 gene compared to the other cerebellar lobes. Therefore, lobe X requires a reduced quantity of CCP1 protein, being less vulnerable to its deficiency in PCD mice.
- COMPLEMENTARY MANUSCRIPT. Lobe X shows an increased basal expression of heat shock protein 25 (HSP25) and its phosphorylated active form compared to the other lobes. HSP25 is known for its protective properties against different types of cellular stress, and we have demonstrated that its expression also increases in the presence of neuronal damage. Therefore, lobe X is more protected than other cerebellar regions against any type of neural damage.
As the Reviewer can appreciate, the first study refers to a decreased vulnerability of lobe X specifically to the pcd mutation. The second one highlights a physiological particularity of lobe X that confers it an innate protection against neuronal damage. It is important to note that while both findings contribute to the neuro-resistance observed in lobe X of PCD mice, they have distinct underlying causes.
Additionally, we have included a hypothesis concerning a possible evolutionary significance of such innate protection of lobe X.
- Figure 2 (regional gene comparison) contains 24 panels (A to X), which may be excessive. Consider integrating key ages (e.g., P20, P30, P50) or utilizing heatmaps to better illustrate the spatiotemporal trends.
Following the suggestion of the reviewer, we have performed a simplified version of the figure 2, which has also colors congruent with the heat map of the supplementary figure 1. The original figure has been changed to a supplementary one (supplementary figure 2) in order to keep all the information of variations of genes at different ages.
- Figures 5–6 (Western blot images) appear somewhat blurry and lack quantitative data. It is recommended to perform band intensity quantification to support the qPCR data, with clear annotation of the lanes.
As it has been reasoned in our manuscript, we rely on qPCR analyses due to their specificity and sensibility, the Western blot data being merely confirmative, just for ratifying a comparable protein expression to the genetic one. That’s why we have not quantified Western blot results, which, in addition, would not be more than semi-quantitative, as explained in the manuscript. In any case, these confirmative data perfectly fit with our qPCR analyses. Moreover, we have obtained some unexpected results when Western blot bands were overexposed, that is to say, in an extreme situation, and we have also discussed them. Finally, the Journal guidelines encourage authors to include the Western blot bands as unmodified as possible. That’s why we have also support the manuscript with raw Western blot supplementary data, showing uncut bands as much as possible.
- The analysis of neuronal survival in vitro and in vivo shows correlations with glutamylation levels. However, it would strengthen the manuscript to include or discuss more direct functional experiments—such as manipulating Ccp1or Ttll1 expression specifically in cerebellar neurons—to establish causality more definitively.
We agree with the Reviewer as more specific experiments would clarify the specific functions of each gene/protein. However, performing those experiments is not a trivial question: it would require necessarily specific models of inducible/constitutive knocking-out for each gene of interest. The generation/acquisition and maintaining of such models is very expensive, especially if several genes are wanted to be analyzed. Furthermore, even having those models in our animal facilities, it is not possible at all to obtain results in the limited time of response required from this journal.
In any case, functional and mechanistic experiments have been already performed by our group and others. Specific patters of de/glutamylation have been analyzed both in PCD mice and in an in vitro KO model for CCP1, which support our results (Muñoz-Castañeda et al., Sci.Rep., 2018). In this sense, the absence of CCP1 in specific moments of postnatal cerebellar development rise as a key factor to cerebellar destructuration.
In summary, this manuscript explores how glutamylation modulates microtubule dynamics and confers neuroprotection in cerebellar neurons, especially in the context of neurodegenerative disorders like PCD. It is suggested that a major revision is necessary before the manuscript can be accepted.
We hope that after implementing the amendments suggested by the Reviewer and answering demands and questions, the new version of the manuscript can be considered suitable for publication.
Reviewer 2 Report
Comments and Suggestions for Authors
The authors' hypothesis is clear, and the data they've presented is compelling. The paper is well-written, and I appreciate how they've integrated their findings with the existing literature.
However, in its current form, I believe the conclusions, while exciting, are still a bit speculative. To truly make this a standout paper, I think the authors need to provide a bit more experimental muscle to support their core claims. Therefore, I recommend a Major Revision and would be happy to re-evaluate the manuscript once these key points are addressed.
Here are my thoughts on what would elevate this paper:
1. Let's Get to the Proteins!
The study's backbone is the qPCR data, which shows significant changes in mRNA levels for key enzymes like CCP1, CCP4, CCP6, and TTLL1. This is a great starting point, but as we know, mRNA doesn't always equal protein. To strengthen the link between gene expression and the proposed mechanism, I would strongly recommend the authors add a few more experiments:
-
Western Blots: The most direct way to confirm their findings would be to perform Western blots to measure the actual protein levels of these key enzymes. Do the protein levels of CCP1, for instance, truly decrease in the same way its mRNA does?
-
Immunofluorescence: An even more elegant approach could be to use immunofluorescence to visualize the subcellular localization of these enzymes. Are they where we'd expect them to be to have this effect?
2. Direct Proof of Glutamylation is Needed
The authors propose that the neuroresistance of Lobe X is due to a "more dynamic, hyperglutamylated cytoskeleton," and this is the most exciting part of the paper. However, this is a hypothesis based on the expression data, not on direct evidence of the post-translational modification itself. To make this a definitive conclusion, I would ask the authors to consider the following:
-
Immunostaining for Glutamylation: They could perform immunofluorescence staining using specific antibodies that recognize glutamylated tubulin. This would provide direct visual proof of their hypothesis by showing differences in the glutamylation state between Lobe X and the other lobes.
-
Biochemical Assays: Alternatively, or in addition, they could perform biochemical assays to quantify the total levels of glutamylated tubulin. This would provide a robust, quantitative measure to support their qualitative claims.
-
Control Experiment: A simple but crucial control would be to treat the cells with a known inhibitor of tubulin glutamylation to see if this reverses the neuroresistance phenotype. This would provide strong evidence for a causal link.
3. Clarifying the Sibling Paper
The repeated reference to "Hernández-Pérez et al., submitted to this same journal" is a bit confusing for a reviewer. It's great that the authors have a comprehensive body of work, but each paper should stand on its own as much as possible. I'd suggest the authors:
-
Provide a preprint link: If the other paper is available as a preprint, a reference to that would be very helpful.
-
Better explanation: If a preprint isn't available, they should clearly explain in the introduction and discussion how the two papers are complementary, and precisely what unique findings are presented in this current manuscript.
4. Exploring the Causal Link
The correlation between low Ccp1 expression and neuroresistance is strong, but correlation is not causation. I would challenge the authors to propose a future experiment (even if they can't do it for this revision) that would directly test the causal link.
-
Gene Knockdown/Overexpression: What happens if they specifically knock down Ccp1 in a different lobe? Does it then become neuroresistant? Conversely, what happens if they overexpress Ccp1 in Lobe X? Does it lose its resistance? This type of experiment would provide powerful evidence for a causal relationship.
This paper has the potential to be a significant contribution to the field. By addressing these points, the authors will move their work from an intriguing hypothesis to a solid, conclusive study. I look forward to seeing the revised version.
Author Response
Response to REVIEWER #2 comments
Thank you very much for your thorough review and helpful suggestions. Certainly, they have improved the quality of our manuscript. Below, you will find a detailed response addressing point-by-point each of your interesting comments and suggestions. In the revised manuscript file, the in-text changes corresponding to your comments are indicated in blue.
The authors' hypothesis is clear, and the data they've presented is compelling. The paper is well-written, and I appreciate how they've integrated their findings with the existing literature.
However, in its current form, I believe the conclusions, while exciting, are still a bit speculative. To truly make this a standout paper, I think the authors need to provide a bit more experimental muscle to support their core claims. Therefore, I recommend a Major Revision and would be happy to re-evaluate the manuscript once these key points are addressed.
Here are my thoughts on what would elevate this paper:
- Let's Get to the Proteins!
The study's backbone is the qPCR data, which shows significant changes in mRNA levels for key enzymes like CCP1, CCP4, CCP6, and TTLL1. This is a great starting point, but as we know, mRNA doesn't always equal protein. To strengthen the link between gene expression and the proposed mechanism, I would strongly recommend the authors add a few more experiments:
- Western Blots:The most direct way to confirm their findings would be to perform Western blots to measure the actual protein levels of these key enzymes. Do the protein levels of CCP1, for instance, truly decrease in the same way its mRNA does?
We rely on qPCR analyses due to their specificity and sensibility, but we completely agree with the reviewer in terms of mRNA expression doesn't always correspond to protein levels. That’s why we performed complementary Western blot analyses after qPCR. As it has been reasoned in our manuscript, the Western blot data were confirmative, just for ratifying a comparable protein expression to the genetic one. In any case, these experiments perfectly corroborated our qPCR analyses, showing the same protein variations that the genetic ones. Here it’s necessary comment that in just one case we have obtained unexpected results: CCP1 protein levels in PCD mice were detectable, whereas Ccp1 mRNA levels are not evident. This result appeared only when Western blot bands were overexposed, that is to say, in an extreme situation, but, in any case, we have discussed them.
- Immunofluorescence:An even more elegant approach could be to use immunofluorescence to visualize the subcellular localization of these enzymes. Are they where we'd expect them to be to have this effect?
Indeed, this has been one of our main concerns! Our first idea before performing Western blot analyses was complementing qPCR analyses with immunohistochemistry. However, we have essayed many different types of antibodies and variables of the technique for the different proteins of interest, and almost no one worked. Our main focus was with CCP1 we have purchased monoclonal and polyclonal commercial antibodies, tried to develop our own antibodies in an associated enterprise (three times), applied different techniques of antigen retrieval, changed immunohistochemistry conditions, essayed many variables of the technique and its developing (fluorescence, diaminobencidine, streptavidine or intensifications), etc. It never worked. Some similar happened with CCP6 and TTLL1. The only antibodies that worked for immunohistochemistry were those directed to CCP4 and, of course, outside the cerebellum, due to its scarcity in this structure. That’s why we decided to use Western blot technique that, even being less specific than immunohistochemistry, at least worked and supported our qPCR results.
- Direct Proof of Glutamylation is Needed
The authors propose that the neuroresistance of Lobe X is due to a "more dynamic, hyperglutamylated cytoskeleton," and this is the most exciting part of the paper. However, this is a hypothesis based on the expression data, not on direct evidence of the post-translational modification itself. To make this a definitive conclusion, I would ask the authors to consider the following:
- Immunostaining for Glutamylation:They could perform immunofluorescence staining using specific antibodies that recognize glutamylated tubulin. This would provide direct visual proof of their hypothesis by showing differences in the glutamylation state between Lobe X and the other lobes.
The experiments of glutamylation concerning CCP1 or other related enzymes (CCP4, CCP6 or even CCP5) already exist. Such experiments demonstrated that a state of hyperglutamylation increases the neuronal vulnerability and predisposes cells to die (Rogowski et al., Cell, 2010). Such experiments were also performed in the PCD model, demonstrating a progressive increase of glutamylation in the cerebellum. Moreover, when such hyperglutamylation was prevented (by downregulating TTLL1), Purkinje cell survival increased. In this case, the particular case of lobe X is not discussed, although it is plausible to think an inverse relationship of the levels of glutamylation and the resistance of cerebellar regions, as we suggested in our manuscript.
In any case, we agree with the reviewer and we have ordered an antibody against glutamylated tubulin. Indeed, in a collaboration with another research unit we are investigating other post-translational modification of tubulins in PCD mice (not only glutamylation but also tyrosination and acetylation), and this work is being prepared for submission to a scientific journal. Briefly, our results demonstrate that the lack of Ccp1 expression induces an increase of glutamylation in the cerebellum of PCD mice. It also causes variations in the distribution pattern of non-tyrosinable form of a-tubulin in the cerebellum together with a parallel loss of expression of b-III-tubulin. However, the absence of Ccp1 was not accompanied either by an accumulation of the tyrosinated state or by a reduction in the detyrosinated forms of a-tubulin in the Purkinje cells. Interestingly, the acetylation of a-tubulin was not modified in the cerebellum of the mutant mice. These results reveal that the Ccp1 gene is directly involved in microtubule dynamics, and suggest that its lack derives on a cytoskeleton destabilization, which may yield on specific neuronal degeneration. Thus, the experiment proposed will fit perfectly with this new work, comparing lobe X and other lobes as two different cerebellar scenarios, both in wild-type and PCD mice. We provide a caption of a figure of this work demonstrating the higher levels of glutamylation in PCD mice (attached PDF). Please, be confidential with this information.
- Biochemical Assays:Alternatively, or in addition, they could perform biochemical assays to quantify the total levels of glutamylated tubulin. This would provide a robust, quantitative measure to support their qualitative claims.
- Control Experiment:A simple but crucial control would be to treat the cells with a known inhibitor of tubulin glutamylation to see if this reverses the neuroresistance phenotype. This would provide strong evidence for a causal link.
Both questions have the same answer. We agree with the Reviewer, as either a biochemical assay or an experiment of inhibition would clarify the function of glutamylation. However, those experiments would require necessarily some time. Unfortunately, it is not possible at all to obtain results in the limited time of response required from this journal. Note that all the tissue of our animals was used for this work. The lobe X of a mouse is very small (especially around P20) and we have used a technique directed to gather and purify RNA specifically, without taking into account proteins (i.e. a separation method with Trizol, even maintaining proteins, is not as suitable for gathering RNA than other specific kits). We would need to process more mice at different ages to achieve this objective.
Concerning the second proposal, precisely, the inhibition of glutamylation would increase the neurorresistance (the “problem” is the hyperglutamylation) and it has been already done (Rogowski et al., Cell, 2010) although not specifically in lobe X. In any case, increasing glutamylation to reverse neuroprotection of lobe X in PCD mice would require to stablish new parental crossings and wait until obtaining mutant animals.
- Clarifying the Sibling Paper
The repeated reference to "Hernández-Pérez et al., submitted to this same journal" is a bit confusing for a reviewer. It's great that the authors have a comprehensive body of work, but each paper should stand on its own as much as possible. I'd suggest the authors:
- Provide a preprint link:If the other paper is available as a preprint, a reference to that would be very helpful.
This is a very good idea, but, unfortunately, we have not done preprints of this second work. Of course, if finally both investigations are published, crossed references will be provided in both scientific papers. Of course, if the Editorial Team of this journal considers it suitable, a direct link could be also provided.
- Better explanation:If a preprint isn't available, they should clearly explain in the introduction and discussion how the two papers are complementary, and precisely what unique findings are presented in this current manuscript.
The Reviewer is right. We have explained the results and conclusions of the complementary work in the new version of the manuscript, as well as the reasons for splitting them. Please, note that other Reviewer had a similar suggestion and some of these changes can be implemented in a color different to blue. In addition, below, we would like to provide to this Referee an explanation of his/her request.
During our investigations, we realized that lobe X of the cerebellum is a neuro-resistant region, PCD mice presenting delayed/attenuated neuronal loss in this region. Intriguingly, such resistance is not exclusive of this model of CONDCA, but practically all the known models of cerebellar damage, either genetic or acquired, share it. Then, we analyzed the basis of such neuro-resistance in our PCD model, and we have reached two different but complementary outcomes, as it is suggested in the manuscript. This is the reason why we have divided this research in two manuscripts. However, the Reviewer is right, as the complementary conclusions of the second work are not adequately explained in this manuscript, but merely mentioned. Below, we summarize the conclusions of both manuscripts. In addition, we have included an expanded version of the results of the second manuscript.
- PRESENT MANUSCRIPT. Lobe X presents a lower basal expression of Ccp1 gene compared to the other cerebellar lobes. Therefore, lobe X requires a reduced quantity of CCP1 protein, being less vulnerable to its deficiency in PCD mice.
- COMPLEMENTARY MANUSCRIPT. Lobe X shows an increased basal expression of heat shock protein 25 (HSP25) and its phosphorylated active form compared to the other lobes. HSP25 is known for its protective properties against different types of cellular stress, and we have demonstrated that its expression also increases in the presence of neuronal damage. Therefore, lobe X is more protected than other cerebellar regions against any type of neural damage.
As the Reviewer can appreciate, the first study refers to a decreased vulnerability of lobe X specifically to the pcd mutation. The second one highlights a physiological particularity of lobe X that confers it an innate protection against neuronal damage. It is important to note that while both findings contribute to the neuro-resistance observed in lobe X of PCD mice, they have distinct underlying causes.
Additionally, we have included a hypothesis concerning a possible evolutionary significance of such innate protection of lobe X.
- Exploring the Causal Link
The correlation between low Ccp1 expression and neuroresistance is strong, but correlation is not causation. I would challenge the authors to propose a future experiment (even if they can't do it for this revision) that would directly test the causal link.
- Gene Knockdown/Overexpression:What happens if they specifically knock down Ccp1 in a different lobe? Does it then become neuroresistant? Conversely, what happens if they overexpress Ccp1 in Lobe X? Does it lose its resistance? This type of experiment would provide powerful evidence for a causal relationship.
We accept and truly appreciate the suggestion of the Reviewer! In any case, performing those experiments is not a trivial question: it would require necessarily specific models of inducible/constitutive knocking-out for Ccp1 in specific regions. The generation/acquisition and maintaining of such models is very expensive, especially if several situations are wanted to be analyzed. In any case, we have collaborated previously with the group of Dr. Annie Andrieux and Dr. MarieJo Moutain from the Université Grenoble-Alpes. They have inducible models similar to PCD mice that may ease the consecution of these experiments. We are sure that they will collaborate with us again to perform the interesting analyses suggested by the Reviewer.
This paper has the potential to be a significant contribution to the field. By addressing these points, the authors will move their work from an intriguing hypothesis to a solid, conclusive study. I look forward to seeing the revised version.
We hope that after implementing the amendments suggested by the Reviewer and answering demands and questions, the new version of the manuscript can be considered suitable for publication.

Reviewer 3 Report
Comments and Suggestions for Authors
The study investigates the molecular mechanisms that make cerebellar lobe X more resistant to neurodegeneration compared to other lobes, using the PCD (Purkinje Cell Degeneration) mouse model, which mimics a human genetic disorder known as CONDCA (Childhood-Onset Neurodegeneration with Cerebellar Atrophy).
This is a very interesting work, and I would therefore like to offer a few minor suggestions to the authors.
Introduction
The introduction is rather long and could be streamlined by reducing repetition of key concepts. Some sentences are overly descriptive and narrative in style.
Materials and Methods
-
The number of animals is relatively small for some analyses (N=5 per group); this limitation should be discussed.
-
The statistical approach is not always well justified: for instance, why was the Kruskal-Wallis test used instead of ANOVA in certain conditions?
-
It is not entirely clear how variability between biological and technical qPCR replicates was controlled.
Results
The results are well organized and consistent with the stated objectives. The figures (especially Fig. 1 and Fig. 2) are clear and informative, and the data are well presented. However, the following points should be improved:
-
CT values >30 for Ccp4 should be interpreted with greater caution: although statistically significant, their reliability is low. Strong conclusions based on these data should be avoided.
-
The interpretation of Ccp6 “compensation” is interesting but lacks functional validation; the speculative nature of this claim should be emphasized.
-
Some parts of the descriptive text (e.g., pages 6–8) are redundant with respect to the figure legends.
Discussion
The discussion is overly lengthy (exceeding 5 pages) and contains several repetitions. Certain sections (e.g., those discussing CCP4 in the cornea or CCP6 in bone marrow) are marginal to the main focus and could be shortened.
Author Response
Response to REVIEWER #3 comments
Thank you very much for your thorough review and helpful suggestions. Certainly, they have improved the quality of our manuscript. Below, you will find a detailed response addressing point-by-point each of your interesting comments and suggestions. In the revised manuscript file, the in-text changes corresponding to your comments are indicated in golden yellow.
The study investigates the molecular mechanisms that make cerebellar lobe X more resistant to neurodegeneration compared to other lobes, using the PCD (Purkinje Cell Degeneration) mouse model, which mimics a human genetic disorder known as CONDCA (Childhood-Onset Neurodegeneration with Cerebellar Atrophy).
This is a very interesting work, and I would therefore like to offer a few minor suggestions to the authors.
Introduction
The introduction is rather long and could be streamlined by reducing repetition of key concepts. Some sentences are overly descriptive and narrative in style.
Following the suggestion of the Reviewer, we have restructured the introduction. In any case, note that the other reviewers have asked for additional information in this section, which may result in an enlargement of some parts.
Materials and Methods
- The number of animals is relatively small for some analyses (N=5 per group); this limitation should be discussed.
The number of animals was limited to 5-7 due to the restrictions of the Bioethics Committee of the University of Salamanca (reference numbers 291 and 613). Following the advices of the 3Rs guidelines for animal research, we had to find an equilibrium between avoiding a waste of animals and having enough statistical power. The number of animals employed here perfectly fit to this equilibrium. In addition, the number of PCD mice in each offspring is a quarter of the total pups. Moreover, they are somehow more delicate that their siblings and their mortality is higher. Therefore, it’s not so easy to obtain high numbers of mutant mice, and 5-7 animals per experimental group is a number coherent with PCD production. Of course, we maintained similar numbers for wild-type animals.
Taking into account the commentary of the Referee, we have discussed the limited numbers of mice in the Material and Methods section.
- The statistical approach is not always well justified: for instance, why was the Kruskal-Wallis test used instead of ANOVA in certain conditions?
We have employed non-parametric statistical tests (i.e. Mann Whitney’s U test and Kruskal-Wallis test) due to the number of animals. Parametrical tests should be employed when samples have a normal distribution and equal variances. Moreover, the normality of samples is closely related with their size, and this is the last criterion for applying parametrical tests. In our case, not all samples in all experiments were normal, and all of them were small. We wanted to be both coherent and restrictive with the applied statistics, and therefore we used always non-parametric tests.
We have included a justification of the employed statistical tests in the new version of the manuscript.
- It is not entirely clear how variability between biological and technical qPCR replicates was controlled.
We have used triplicates for our samples in all qPCR experiments to reduce or, at least, control the variability of these experiments. Moreover, the device employed (the thermocycler QuantStudioTM 7 Flex Real-Time PCR System) takes into account these replicates and automatically analyze them. Just when variability between replicates is small, the result is considered valid. If it not were the case, the program alerts the researchers to analyze in particular the problematic triplicate and take a decision: discard the most deviated replicate or repeat the measurement of this sample. We have synthesized and included this idea in the new version of the manuscript.
Results
The results are well organized and consistent with the stated objectives. The figures (especially Fig. 1 and Fig. 2) are clear and informative, and the data are well presented. However, the following points should be improved:
- CT values >30 for Ccp4should be interpreted with greater caution: although statistically significant, their reliability is low. Strong conclusions based on these data should be avoided.
We entirely agree with the Reviewer. That’s why we have clarified in the Results section that data concerning Ccp4 must be considered with caution. We have also included this warning in the Discussion to remember readers that, even being coherent, results derived from the expression of this gene may be biased. Moreover, we have shortened the Discussion (see your final suggestion, later), especially those less important points that may be related with Ccp4. In any case, no strong conclusions based on these data are provided.
- The interpretation of Ccp6“compensation” is interesting but lacks functional validation; the speculative nature of this claim should be emphasized.
Following the advice of the Referee, we have declared that this interpretation is just a hypothesis in the new version of the manuscript.
- Some parts of the descriptive text (e.g., pages 6–8) are redundant with respect to the figure legends.
The legend of figure 1 has been synthesized to avoid redundancies with the main text, as suggested by this Reviewer. We have changed figure 2 as another Referee requested. Then, the figure legend has change completely. In any case, the ancient figure 2 is maintained as supplementary to preserve its original information, and since this Reviewer likes it (we sincerely appreciated that). The other figure legends do not seem redundant with text, especially after the amendments done in the new version of the manuscript
Discussion
The discussion is overly lengthy (exceeding 5 pages) and contains several repetitions. Certain sections (e.g., those discussing CCP4 in the cornea or CCP6 in bone marrow) are marginal to the main focus and could be shortened.
Following the advices of the Referee, we have shortened the discussion. In any case, note that the other reviewers have asked for additional information in this section, which may result in an enlargement of some parts.
We hope that after implementing the amendments suggested by the Reviewer and answering demands and questions, the new version of the manuscript can be considered suitable for publication.